# Quantification of human mature frataxin protein expression in nonhuman primate hearts after gene therapy

Teerapat Rojsajjakul[1], Juliette J. Hordeaux[2], Gourav R. Choudhury[2], Christian J. Hinderer[2], Clementina Mesaros[1], James M. Wilson [2✉] & Ian A. Blair [1✉]

Deficiency in human mature frataxin (hFXN-M) protein is responsible for the devastating neurodegenerative and cardiodegenerative disease of Friedreich's ataxia (FRDA). It results primarily through epigenetic silencing of the *FXN* gene by GAA triplet repeats on intron 1 of both alleles. GAA repeat lengths are most commonly between 600 and 1200 but can reach 1700. A subset of approximately 3% of FRDA patients have GAA repeats on one allele and a mutation on the other. FRDA patients die most commonly in their 30s from heart disease. Therefore, increasing expression of heart hFXN-M using gene therapy offers a way to prevent early mortality in FRDA. We used rhesus macaque monkeys to test the pharmacology of an adeno-associated virus (AAV)hu68.CB7.hFXN therapy. The advantage of using non-human primates for hFXN-M gene therapy studies is that hFXN-M and monkey FXN-M (mFXN-M) are 98.5% identical, which limits potential immunologic side-effects. However, this presented a formidable bioanalytical challenge in quantification of proteins with almost identical sequences. This could be overcome by the development of a species-specific quantitative mass spectrometry-based method, which has revealed for the first time, robust transgene-specific human protein expression in monkey heart tissue. The dose response is non-linear resulting in a ten-fold increase in monkey heart hFXN-M protein expression with only a three-fold increase in dose of the vector.

[1] Penn/CHOP Friedreich's Ataxia Center of Excellence and Department of Systems Pharmacology and Translational Therapeutics, Perelman School of Medicine, University of Pennsylvania, Philadelphia, PA 19104, USA. [2] Gene Therapy Program, Departments of Medicine and Pediatrics, Perelman School of Medicine, University of Pennsylvania, Philadelphia, PA 19104, USA. ✉email: wilsonjm@upenn.edu; ianblair@upenn.edu

Friedreich's ataxia (FRDA) is a neurodegenerative and cardiodegenerative autosomal recessive genetic disease resulting from an intronic GAA triplet repeat expansion in the *FXN* gene[1,2]. FRDA has a prevalence of 1 in 50,000–100,000 individuals in the USA, and so is the most common hereditary ataxia[3]. This devastating disease, which is characterized by ataxia and other neurological defects[3], arises from a deficiency in human mature frataxin (hFXN-M), a 130 amino acid mitochondrial protein[4]. Full-length hFXN protein (1–210, Fig. 1) expressed in the cytosol, translocates to the mitochondria where it undergoes sequential mitochondrial processing peptidase (MPP) cleavage at $G^{41}$-$L^{42}$ and $K^{80}$-$S^{81}$ to produce hFXN-M protein (Fig. 1)[5,6]. Mitochondrial hFXN-M protein plays an important role in the biogenesis and maintenance of Fe-S clusters and in persulfide processing[7–10]. Epigenetic silencing of the *FXN* gene due to the presence of GAA triplet repeats in intron 1 of both alleles of the *FXN* gene (homozygous patients) results in reduced transcription of *FXN* mRNA, reduced expression of full-length hFXN (1–210) in the cytosol, and reduced amounts of hFXN-M produced in the mitochondria[1,11]. GAA repeat lengths are most commonly between 600 and 1200 in FRDA patients[1], although a repeat length of 1700 has been reported in one patient[12]. A subset of approximately 3% of FRDA patients have a mutation on one allele and GAA repeats in intron 1 of the other allele (complex heterozygotes)[1,4]. The presence of mutated proteins that could potentially be expressed in complex heterozygous patients has not been reported, suggesting that the small amount of transcribed hFXN-M protein comes from the allele containing GAA repeats in intron 1. Symptoms of FRDA generally appear during adolescence; patients slowly progress to wheelchair dependency within 15 years and die most commonly in their 30s from heart disease[3]. Expression of hFXN-M decreases with increased GAA repeat length causing an earlier age of onset and increased disease severity[13,14]. Therefore, increasing expression of hFXN-M in the heart of both homozygous and complex heterozygous patients using gene therapy offers a potential approach to prevent early mortality resulting from cardiac failure.

Studies on gene therapy using rodent models of FRDA have provided encouragement for this approach as well as highlighting the need for caution[15]. For gene delivery, the most common approach has been the use of adeno-associated viruses (AAVs) because the gene delivered by this vector does not integrate into the patient genome and has a low immunogenicity[16]. In addition, the potential of these vectors has been established by numerous preclinical and clinical studies, as well as by already approved therapies[17,18]. Intravenous systemic delivery of the gene is the most widely used method because of its extremely low invasiveness[15]. This approach was used in the conditional Mck mouse model of FRDA where there is complete deletion of the *FXN* gene in cardiac and skeletal muscle[19]. Intravenous administration by the retro-orbital route with AAVrh10 expressing hFXN-M (5.4E13 vector genomes/kg) resulted in the vector being readily transported to the myocardium where hFXN-M expression prevented the onset of cardiac disease. Furthermore, later administration of the same amount of AAVrh10 vector after the onset of heart failure, was able to completely reverse the cardiomyopathy of the mice at the functional, cellular, and molecular levels[19]. A subsequent study was conducted in a mouse model with partial cardiac-specific excision of *FXN* exon 4 in the heart using Cre-Lox recombination[20]. This FRDA cardiac-specific mouse model has a mild phenotype like the early human clinical cardiomyopathy of FRDA where the clinical cardiac phenotype requires stress. A single intravenous administration of AAVrh.10hFXN (1.00E11 vector genomes) was found to relieve the phenotypic outcomes of cardiomyopathy in this cardiac-specific FRDA mouse model[20].

It has been reported that hFXN-M cardiac overexpression up to 9-fold the normal endogenous mouse FXN-M proteoform levels in mice was safe, but significant toxicity to the heart at levels above 20-fold was found[21]. However, the methodology that was used did not establish whether the hFXN-M was truncated in the mouse heart, or whether it was present in the cytosol as well as the mitochondria, like mouse FXN-M[22]. Therefore, we reasoned that further studies were required to firmly establish the relationship between the dose of the hFXN-M vector and the expression of intact hFXN-M protein in the mitochondria of target tissue. To address this important issue, we used rhesus macaque monkeys as a nonhuman primate model because

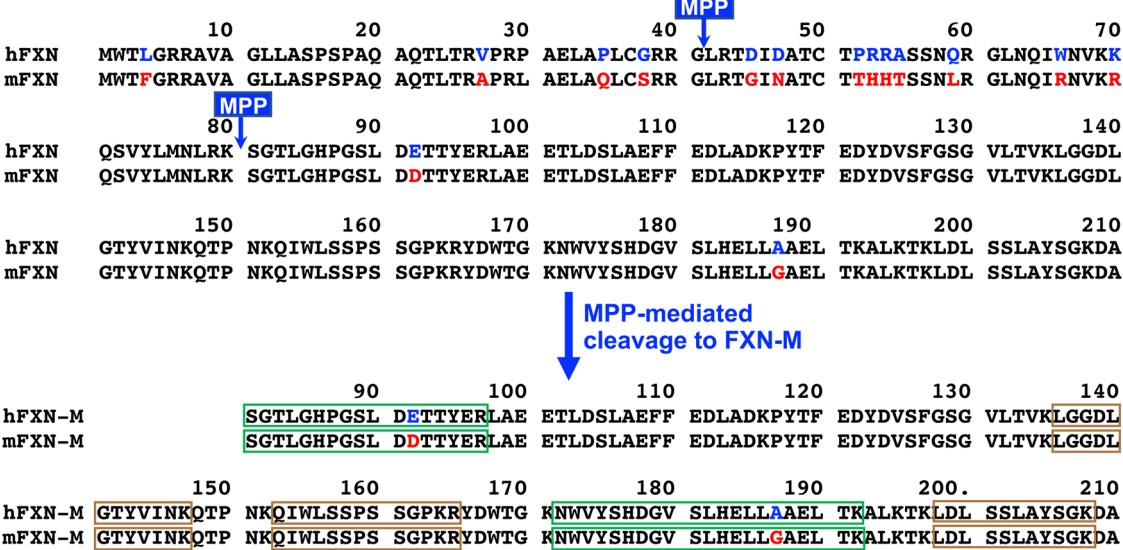

**Fig. 1 Formation of hFXN-M (81–210, upper) and Resus macaque mFXN-M (lower, 81–210).** Full-length hFXN (1–210) and full-length mFXN (1–210) translocate to the mitochondria where both hFXN-M (81–210, upper) and mFXN-M (81–210, lower) are formed as 130 amino acid proteins by sequential mitochondrial processing peptidase (MPP)-mediated cleavage at $G^{41}$-$L^{42}$ and $K^{80}$-$S^{81}$. Amino acid differences are shown in blue (human) and red (monkey). The two tryptic peptides that were quantified to differentiate hFXN-M from mFXN-M are shown in green boxes, whereas three common tryptic peptides that were analyzed as qualifying peptides are shown in brown boxes.

hFXN-M and monkey FXN-M (mFXN-M) are 98.5% identical and both are formed in mitochondria from full-length FXN protein in a similar manner (Fig. 1). Using a highly specific stable isotope dilution immunoprecipitation ultra-high performance liquid chromatography-multiple reaction monitoring mass spectrometry (IP-UHPLC-MRM/MS) method, we have quantified both hFXN-M and mFXN-M in Rhesus macaque monkeys 1-month after they were treated with escalating doses of an AAVhu68 clade F vector expressing hFXN-M under the ubiquitous CB7 promoter.

## Results

**IP-UHPLC-MRM/MS analysis of control heart tissues.** Bioactive proteins are found in heart tissue in the presence of high abundance proteins (HAPs) such as vimentin and myosin that are often $10^6$ to $10^7$ higher in concentration. Simple extraction procedures result in interference from the HAPs, as well as suppression of the MS signal from the target protein by the HAPs. Removal of the HAPs by immunodepletion can cause the loss of the target protein through non-covalent binding to them[23]. Low abundance proteins such hFXN-M and mFXN-M can also be lost during the extraction of the biofluid or tissue through non-covalent binding to glassware and plastic surfaces as we have shown previously for amyloid-β proteins[24]. Immunoprecipitation (IP) can be used to purify target proteins, but recovery from different heart tissue samples can be inconsistent. Digestion by proteases, such as trypsin, which is required to generate peptides that are amenable to specific and sensitive quantification by UHPLC-MRM/MS (Fig. 2A), can also be inconsistent for different tissue samples. UHPLC-MRM/MS is not a quantitative tool because differential ionization of peptides can occur in the source of the mass spectrometer. Therefore, a heavy isotope internal protein standard prepared using stable isotope labeling by amino acids in cell culture (SILAC), is added to the heart tissue sample at the start of the isolation procedure (Fig. 2A). The SILAC standard acts as a carrier to prevent losses during isolation, as an internal control to normalize extraction and protease digestion efficiency, and to compensate for differential ionization in the mass spectrometer source. This permits accurate and precise protein quantification to be conducted[14]. In the present study, homogenized frozen control (non-FRDA) human and rhesus macaque monkey heart tissues (25 mg to 150 mg) spiked with SILAC-hFXN-M (40 ng) were purified by IP (Fig. 2A). The IP removed most of the interfering proteins and the SILAC-hFXN-M provided an internal control that compensated for any losses during the procedure. Protease digestion of the hFXN-M and mFXN-M in the IP eluate with trypsin provided numerous peptides from each protein (Fig. 3). The identity of hFXN-M (Fig. 3A) and mFXN-M (Fig. 3B) was established by the presence of 5 tryptic peptides from each protein in the relevant UHPLC-MRM/MS chromatogram. The use of three MRM transitions for each of the peptides provided the specificity that made it possible to unequivocally differentiate the two proteins (Table 1). There are two amino acid differences between hFXN-M and mFXN-M (**E**92**D** and **A**187**G**) in tryptic peptides S81GTLGHPGSLD**E**TTYER97, N172WVYSHDGVSL HELL**A**AELTK192 (hFXN-M) and S81GTLGHPGSLD**D**TTYER97,

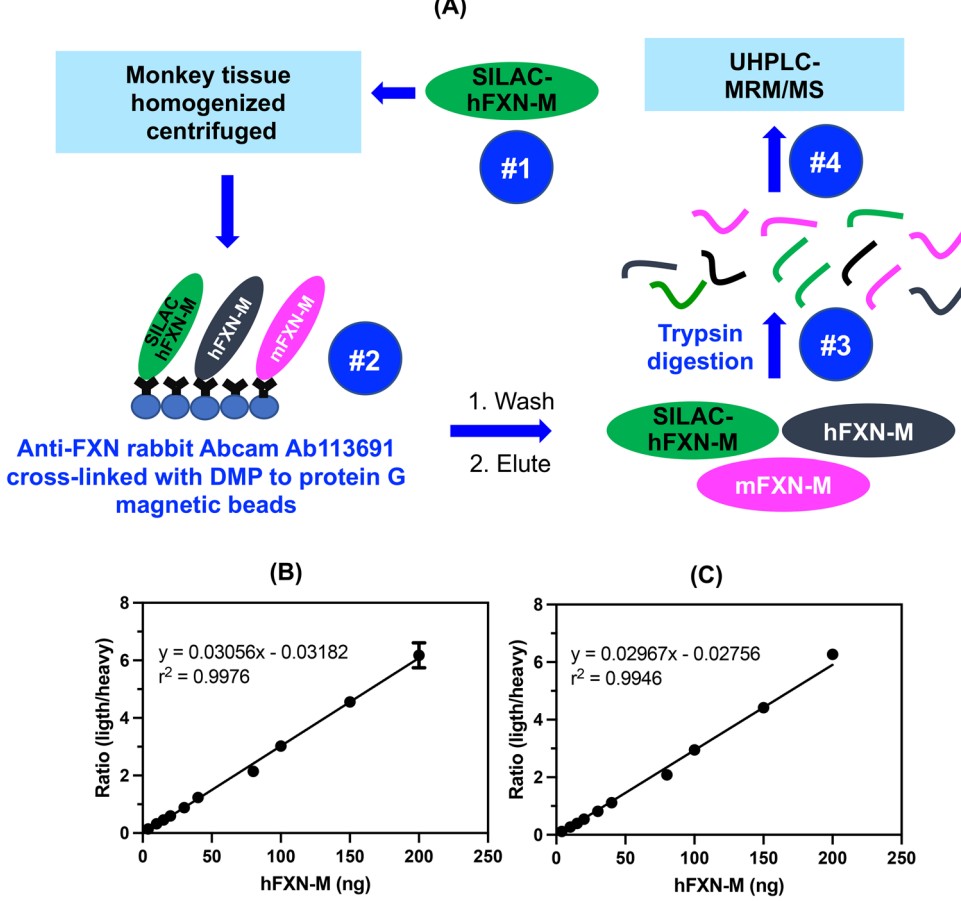

**Fig. 2 Workflow for quantification of hFXN-M and mFXN-M. A** Schematic showing workflow for quantification of hFXN and mFXN-M. **B** Typical standard curve for SGTLGHPGSLD**E**TTYER tryptic peptide used for quantitative analysis of hFXN-M and mFXN-M in monkey heart. **C** Typical standard curve for KNWVYSHDGVSLHELL**A**AELTK tryptic peptide for quantitative analysis of hFXN-M and mFXN-M in monkey heart.

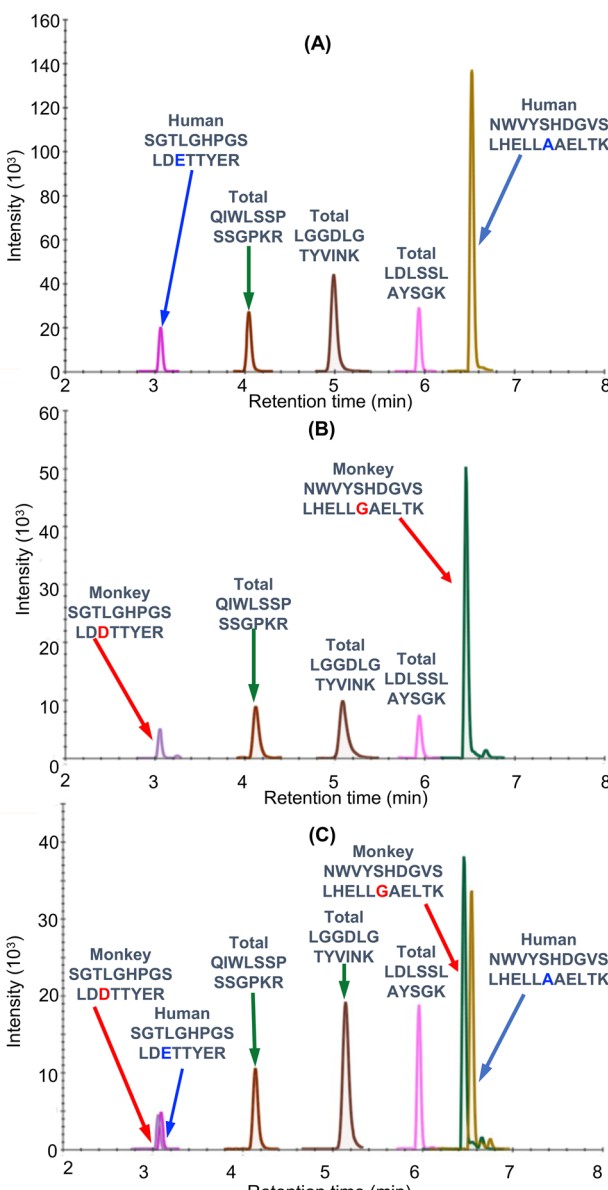

**Fig. 3 UHPLC-MRM/MS analysis of FXN-M tryptic peptides. A** hFXN-M from human heart left ventricle. **B** mFXN-M from Rhesus macaque heart left ventricle. **C** 50:50 mixture of hFXN-M and mFXN-M. The most intense MRM transitions for each peptide were used: SGTLGHPGSLD**D**TTYER ($y_{14}^{2+}$), SGTLGHPGSLD**E**TTYER ($y_{14}^{2+}$), QIWLSSPSSGPKR ($y_8^+$), LGGDLGTYVINK ($y_{10}^+$), LDLSSLAYSGK ($y_{10}^+$), NWVYSHDGVSLHELL**G**AELTK ($y_{19}^{3+}$), NWVYSHDGVSLHELL**A**AELTK ($y_{19}^{3+}$).

**Table 1 MRM transitions and UHPLC retention times for analysis of hFXN-M and mFXN-M tryptic peptides.**

| # | Start | End | Peptide | Hu/M | L/H | Parent ion | m/z | Prod ion 1 | m/z | Prod ion 2 | m/z | Prod ion 3 | m/z | RT (min) |
|---|---|---|---|---|---|---|---|---|---|---|---|---|---|---|
| 1 | 81 | 97 | SGTLGHPGSLD**D**TTYER | M | L | MH$_3^{3+}$ | 602.615 | y$_{14}^{2+}$ | 780.868 | y$_5^+$ | 669.320 | y$_3^+$ | 467.225 | 3.03 |
| 2 | 81 | 97 | SGTLGHPGSLD**E**TTYER | Hu | L | MH$_3^{3+}$ | 607.287 | y$_{14}^{2+}$ | 787.876 | y$_5^+$ | 669.320 | y$_3^+$ | 467.225 | 3.07 |
| 3 | 81 | 97 | SGTLGHPGSLD**E**TTYER | Hu | H | MH$_3^{3+}$ | 611.300 | y$_{14}^{2+}$ | 793.896 | y$_5^+$ | 669.320 | y$_4^+$ | 568.273 | 3.07 |
| 4 | 153 | 165 | QIWLSSPSSGPKR | Hu & M | L | MH$_2^{2+}$ | 721.891 | y$_{10}^+$ | 1015.553 | y$_9^+$ | 902.469 | y$_8^+$ | 815.437 | 4.10 |
| 5 | 153 | 165 | QIWLSSPSSGPKR | Hu & M | H | MH$_2^{2+}$ | 724.901 | y$_{10}^+$ | 1021.573 | y$_9^+$ | 902.469 | y$_8^+$ | 815.437 | 4.10 |
| 6 | 136 | 147 | LGGDLGTYVINK | Hu & M | L | MH$_2^{2+}$ | 625.343 | y$_8^+$ | 907.525 | y$_7^+$ | 794.441 | y$_2^+$ | 261.156 | 5.05 |
| 7 | 136 | 147 | **L**GGD**L**GTYVINK | Hu & M | H | MH$_2^{2+}$ | 631.363 | y$_8^+$ | 913.545 | y$_7^+$ | 794.441 | y$_2^+$ | 261.156 | 5.05 |
| 8 | 198 | 208 | **L**D**L**SS**L**AYSGK | Hu & M | L | MH$_2^{2+}$ | 577.309 | y$_8^+$ | 812.415 | y$_6^+$ | 638.351 | y$_5^+$ | 525.267 | 5.90 |
| 9 | 198 | 208 | **L**D**L**SS**L**AYSGK | Hu & M | H | MH$_2^{2+}$ | 586.339 | y$_8^+$ | 818.435 | y$_6^+$ | 644.371 | y$_5^+$ | 525.267 | 5.90 |
| 10 | 172 | 192 | N**W**VYSHDGVSLHELL**G**AELTK | M | L | MH$_4^{4+}$ | 592.804 | y$_{19}^{3+}$ | 690.029 | y$_8^+$ | 844.514 | y$_7^+$ | 731.430 | 6.46 |
| 11 | 172 | 192 | NWVYSHDGVSLHELL**A**AELTK | Hu | L | MH$_4^{4+}$ | 596.308 | y$_{19}^{3+}$ | 694.700 | y$_8^+$ | 858.530 | y$_7^+$ | 745.445 | 6.56 |
| 12 | 172 | 192 | NWVYSHDGVSL**H**ELL**A**AELTK | Hu | H | MH$_4^{4+}$ | 602.328 | y$_{19}^{3+}$ | 702.727 | y$_8^+$ | 876.590 | y$_7^+$ | 757.486 | 6.56 |

Hu = human, M = monkey, L = light peptide, H = heavy peptide, **D** and **G** are mFXN-M amino acids, **E** and **A** are hFXN-M amino acids, **L** = [$^{13}C_6$]-leucine; Prod ion = Product ion.; RT = UHPLC retention time; m/z = mass to charge ratio.

N$^{172}$WVYSHDGVSLHELL**G**AELTK$^{192}$ (mFXN-M) (Fig. 1). Therefore, 7 tryptic peptides were observed in the UHPLC-MRM/MS chromatogram when a 50:50 mixture of the two proteins was digested with trypsin (Fig. 3C).

Standard curves prepared in bovine serum albumin surrogate matrix that had been subjected to the same IP-LC-MRM/MS procedure were linear (Fig. 2B, C, Supplementary Data 1). Back-calculated amounts of hFXN-M from standard curve samples ($n = 5$) for SGTLGHPGSLD**E**TTYER (Fig. 2B, Supplementary Data) and NWVYSHDGVSLHELL**A**AELTK (Fig. 2C, Supplementary Data 1) from hFXN-M had a precision of better than 15% and accuracy of between. 85% and 115% (Table 2A, Supplementary Data 2 and Table 2B, Supplementary Data 3). UHPLC-MRM/MS was conducted using the transitions for the light tryptic (unlabeled)

**Table 2 Back-calculated amounts of hFXN-M from standard curve samples ($n = 5$).**

| (A) | | | | | (B) | | | | |
|---|---|---|---|---|---|---|---|---|---|
| Theoretical (ng) | Calculated (ng) | SD (ng) | CV (%) | Accuracy (%) | Theoretical (ng) | Calculated (ng) | SD (ng) | CV (%) | Accuracy (%) |
| 4.0 | 4.3 | 0.57 | 13% | 108% | 4.00 | 3.83 | 0.24 | 6% | 96% |
| 10.0 | 10.3 | 1.57 | 15% | 103% | 10.00 | 9.05 | 0.65 | 7% | 90% |
| 15.0 | 14.8 | 1.01 | 7% | 98% | 15.00 | 13.49 | 0.50 | 4% | 90% |
| 20.0 | 19.5 | 1.56 | 8% | 97% | 20.00 | 18.35 | 0.50 | 3% | 92% |
| 30.0 | 29.1 | 2.31 | 8% | 97% | 30.00 | 27.72 | 1.49 | 5% | 92% |
| 40.0 | 40.8 | 2.50 | 6% | 102% | 40.00 | 38.16 | 0.51 | 1% | 95% |
| 80.0 | 71.4 | 1.47 | 2% | 89% | 80.00 | 71.19 | 1.03 | 1% | 89% |
| 100.0 | 100.8 | 6.58 | 7% | 101% | 100.00 | 100.98 | 2.79 | 3% | 101% |
| 150.0 | 152.0 | 3.17 | 2% | 101% | 150.00 | 151.19 | 4.45 | 3% | 101% |
| 200.0 | 206.1 | 14.46 | 7% | 103% | 200.00 | 214.86 | 3.32 | 2% | 107% |

(A) SGTLGHPGSLDETTYER.
(B) NWVYSHDGVSLHELLAAELTK.

peptides shown in Table 1. A chromatogram of the most abundant UHPLC-MRM/MS tryptic peptide transitions from human heart revealed an excellent separation of the five peptides from hFXN-M. The amino terminal SGTLGHPGSLD**E**TTYER tryptic peptide, which distinguished hFXN-M from mFXN-M by virtue of **E**-92 instead of **D**-92, was the most polar peptide and eluted at 3.05-min (Fig. 3A). The least polar peptide NWVYSHDGVSLHELL**A**AELT-KALK, which distinguished hFXN-M from mFXN-M by virtue of a **A**-187 instead of **G**-187, eluted at 6.56-min. A chromatogram of the most abundant UHPLC-MRM/MS transitions of mFXN tryptic peptides from monkey heart also revealed that there was excellent separation of the five tryptic peptides from mFXN-M (Fig. 3B). The N-terminal SGTLGHPGSLD**D**TTYER tryptic peptide, which was the most polar peptide, eluted at 3.03-min. The least polar peptide NWVYSHDGVSLHELL**G**AELTKALK, eluted at 6.44-min (Fig. 3B). A chromatogram from a 50:50 mixture of hFXN-M from human heart and mFXN-M from monkey heart showed that the two tryptic peptides with different amino acid sequences (SGTLGH PGSLD**E**TTYER and SGTLGHPGSLD**D**TTYER, NWVYSHDG VSLHELL**A**AELTKALK and NWVYSHDGVSLHELL**G**AELT-KALK) could be separated from each other so that hFXN-M and mFXN-M proteins could be readily distinguished (Fig. 3C). The two unique tryptic peptides from hFXN and mFXN were separated either through their different MRM transitions (N-terminal SGT peptides) or by a combination of their different MRM transitions and different retention times (NWV-peptides) (Table 1).

**Stable isotope dilution IP-UHPLC-MRM/MS quantitative analysis of FXN-M in control (non-FRDA) heart tissues.** Many quantitative MS-based studies of protein expression rely on the use of isotopically stable isotope-labeled peptide (AQUA) standards. AQUA standards give excellent precision because they compensate for differences in the ionization efficiency in the mass spectrometer. However, they have poor accuracy, particularly when IP is used for protein isolation because they do not take account of losses during the procedure. They also do not take account of inter-sample differences in the efficiency of protein digestion as we showed for apolipoprotein (Apo)A1 protein in human serum[25]. Addition of an AQUA peptide prior to protease digestion results in differential loss of the peptide during diges-tion when compared with the protein-derived peptide[26] and so this approach cannot be used. These problems can be readily overcome using SILAC protein internal standards as we have demonstrated for amyloid-β proteins in cerebrospinal fluid (CSF)[24], ApoA1 in serum[25], hFXN-M and hFXN-E in whole blood[14], high mobility group box1 (HMGB1) in human cells[27], plasma[28], and serum[28], oxidized HMGB1 in cell media[29], and

mouse FXN-M in mouse heart, brain, and liver tissues[22]. The ratio between the endogenous protein and the SILAC protein internal standard is established at the start of the isolation pro-cedure. This ratio remains the same throughout the entire pro-cedure and is used to calculate the amount of endogenous protein from a standard curve that is constructed at the same time with an authentic protein standard. The SILAC protein also serves as a carrier to enhance the recovery of low-level tissue proteins that can be lost through non-selective binding to glassware and plastic surfaces. This was unequivocally demonstrated in our assay for amyloid-β proteins in CSF[24], where the proteins are almost completely lost in the absence of a stable isotope carrier through binding to surfaces during isolation and analysis. Therefore, we employed SILAC-hFXN-M as the internal standard for quanti-fying both hFXN-M and mFXN-M in monkey heart.

Typical UHPLC-MRM/MS chromatograms for SGTLGHPGS LD**E**TTYER tryptic peptide from control (non-FRDA) human heart (upper) and heavy SGT**L**GHPGS**L**D**E**TTYER internal standard (lower) are shown in Fig. 4a. Chromatograms for NWVYSHDGVS LHELL**A**AELTK tryptic peptide from control (non-FRDA) human heart (upper) and heavy NWVYSHDGVS**L**HELL**A**AEL**T**K internal standard (lower) are shown in Fig. 4b. The amount of hFXN-M was then determined for each peptide from the relevant standard curve (Fig. 2B, C). The mean level of hFXN-M (4.8 ng/mg tissue) was then calculated from the mean of the two hFXN-M and mFXN-M peptides. Typical UHPLC-MRM/MS chromatograms for SGTLGH PGSLD**D**TTYER tryptic peptide from control monkey heart (upper) and heavy SGT**L**GHPGS**L**D**E**TTYER internal standard (lower) are shown in Fig. 4c. Chromatograms for NWVYSHDG VSLHELL**G**AELTK tryptic peptide from control monkey heart (upper) and heavy NWVYSHDGVS**L**HE**LLA**AE**L**TK internal standard (lower) are shown in Fig. 4d. The ratio of sum of the three light peptide MRM transitions to the sum of the three heavy peptide transitions were determined. The amount of each peptide was then determined for each peptide from the relevant standard curve (Fig. 2B, C). The mean level of mFXN-M (1.9 ng/mg tissue) was then calculated from the mean of the two mFXN-M peptides. Levels of hFXN-M in five different control (non-FRDA) left ventricle human heart tissue were determined to be 5.1 ± 1.7 ng/mg tissue (mean ± standard deviation (SD); $n = 5$). Levels of mFXN-M in five different Rhesus macaque left ventricle heart tissue from untreated animals were 2.1 ± 0.4 ng/mg tissue (Fig. 4e, SupplementaryData 4).

**All AAV vector doses resulted in hFXN-M expression in monkey heart.** UHPLC-MRM/MS chromatograms of monkey tissue after the lowest dose of the AAV vector revealed the presence of the

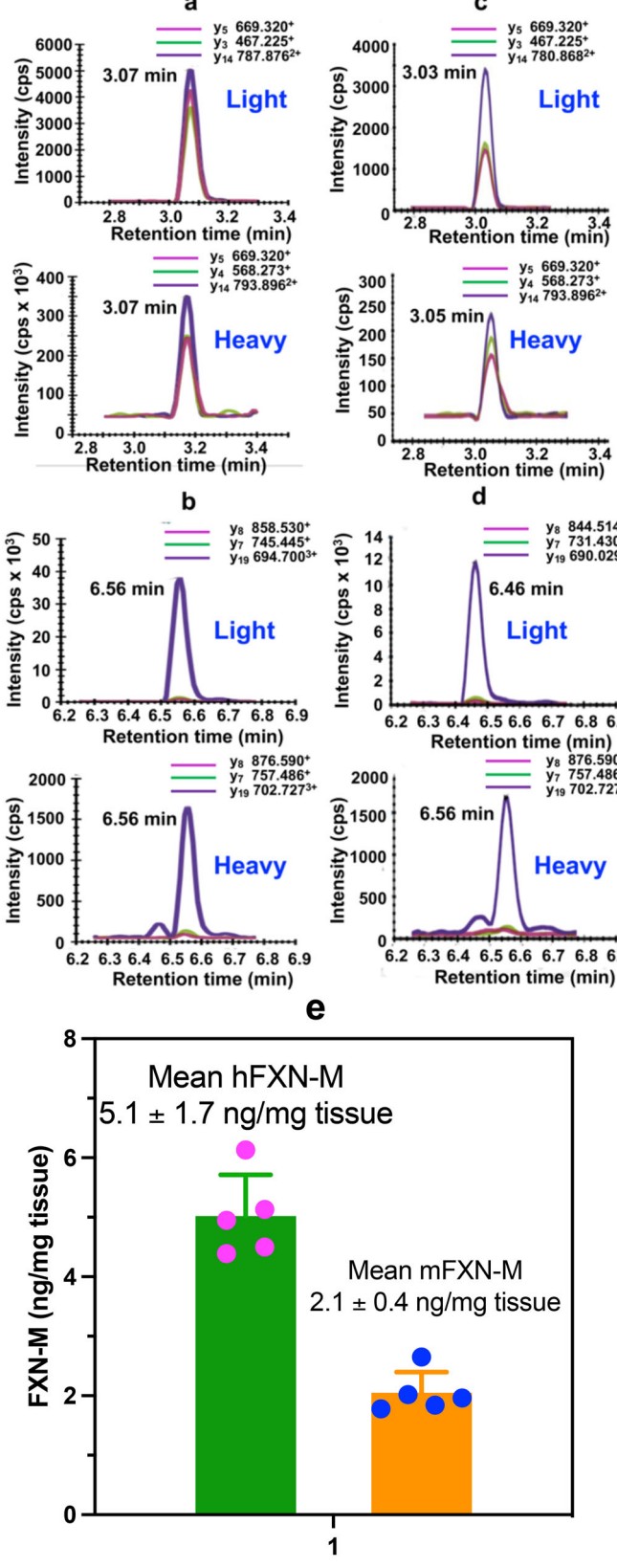

**Fig. 4 Typical UHPLC-MRM/MS chromatograms of tryptic peptides for quantification of hFXN-M (4.8 ng/mg tissue) from a control (non-FRDA) human heart and mFXN-M (1.9 ng/mg tissue) from a control Resus macaque monkey heart. a** SGTLGHPGSLD**E**TTYER tryptic peptide from hFXN-M ($MH_3^{3+}$ = 607.287; upper) and heavy SGT**L**GHPGS**L**D**E**TTYER internal standard ($MH_3^{3+}$ = 611.300; lower). **b** NWVYSHDGVSLHELL**A**AELTK tryptic peptide from hFXN-M ($MH_4^{4+}$ = 596.308; upper) and heavy NWVYSHDGVS**L**HE**LLA**AE**L**TK internal standard ($MH_4^{4+}$ = 602.328; lower). **c** SGTLGHPGSLD**D**TTYER tryptic peptide from mFXN-M ($MH_3^{3+}$ = 602.615; upper) and heavy SGT**L**GHPGS**L**D**E**TTYER internal standard ($MH_3^{3+}$ = 611.300; lower). **d** NWVYSHDGVSLHELL**G**AELTK tryptic peptide from mFXN-M ($MH_4^{4+}$ = 592.804; upper) and heavy NWVYSHDGVS**L**HE**LLA**AE**L**TK internal standard ($MH_4^{4+}$ = 602.328; lower). **e** Levels of hFXN-M in control (non-FRDA) human heart tissue (mean ± SD; *n* = 5) and mFXN-M in Rhesus macaque heart tissue from untreated animals (mean ± SD; *n* = 5). **L** = [$^{13}C_6$]-leucine.

MRM/MS chromatograms of monkey tissue after the middle dose of the AAV vector revealed that the two specific hFXN peptides (SGTLGHPGSLD**E**TTYER and NWVYSHDGVSLHELL**A**AELT-KALK) were present at much higher intensity than two specific mFXN peptides (SGTLGHPGSLD**D**TTYER and NWVYSHDGVSL-HELL**G**AELTKALK) (Fig. 5B). The three common tryptic peptides (VLTVKLGGDLGTYVINK, QIWLSSPSSGPKRYDWTG, and TKALKTKLDLSSLAYSGK) were also detected (Fig. 5B). UHPLC-MRM/MS chromatograms of monkey tissue after the highest dose of the AAV vector (1.00E14 GC/kg) revealed that the two specific hFXN peptides (SGTLGHPGSLD**E**TTYER and NWVYSHDGVSLHEL-L**A**AELTKALK) were extremely intense (Fig. 5C). In contrast, the two specific mFXN peptides (SGTLGHPGSLD**D**TTYER and NWVYSHDGVSLHELL**G**AELTKALK) could barely be detected. The three common tryptic peptides (VLTVKLGGDLGTYVINK, QIWLSSPSSGPKRYDWTG, and TKALKTKLDLSSLAYSGK) were also detected (Fig. 5C).

**hFXN-M and mFXN-M levels in monkey heart were comparable after the lowest AAV dose.** Typical UHPLC-MRM/MS chromatograms of tryptic peptides for quantification of hFXN-M and mFXN-M in monkey heart after the lowest dose of 1.00E13 GC/kg of the AAV vector expressing hFXN-M are shown in Fig. 6. Chromatograms for SGTLGHPGSLD**E**TTYER tryptic peptide from human heart (upper) and SILAC-SGTLGHPGS**L**D**E**TTYER internal standard (lower) are shown in Fig. 6A. Chromatograms for NWVYSHDGVSLHELL**A**AELTK tryptic peptide from human heart (upper) and heavy NWVYSHDGVS**L**HE**LLA**AE**L**TK internal standard (lower) are shown in Fig. 6C. The amount of hFXN-M was then determined for each peptide from the relevant standard curve (Fig. 2B, C). The mean level of hFXN-M for this sample (1.5 ng/mg tissue, monkey # 2, RV2 B7, 180616) was then calculated from the mean of the two hFXN-M peptides. Chromatograms for SGTLGHPGSLD**D**TTYER tryptic peptide from monkey heart (upper) and heavy SGT**L**GHPGS**L**D**E**TTYER internal standard (lower) are shown in Fig. 6B. Chromatograms for NWVYSHDGVSLHELL**G**AELTK tryptic peptide from monkey heart (upper) and heavy NWVYSHDGVS**L**HE**LLA**AE**L**TK internal standard (lower) are shown in Fig. 6D. The ratio of sum of the three light peptide MRM transitions to the sum of the three heavy peptide transitions were determined. The amount of each peptide was then determined for each peptide from the relevant standard curve (Fig. 2B, C). The mean level of mFXN-M for this sample (3.7 ng/mg tissue, monkey # 2, RV2 B7, 180616) was then calculated from the mean of the two mFXN-M peptides.

two specific hFXN peptides (SGTLGHPGSLD**E**TTYER and NWVYSHDGVSLHELL**A**AELTKALK) as well as the two specific mFXN peptides (SGTLGHPGSLD**D**TTYER and NWVYSHDGVSL-HELL**G**AELTKALK) (Fig. 5A). The three common tryptic peptides (VLTVKLGGDLGTYVINK, QIWLSSPSSGPKRYDWTG, and TKALKTKLDLSSLAYSGK) were also detected (Fig. 5A). UHPLC-

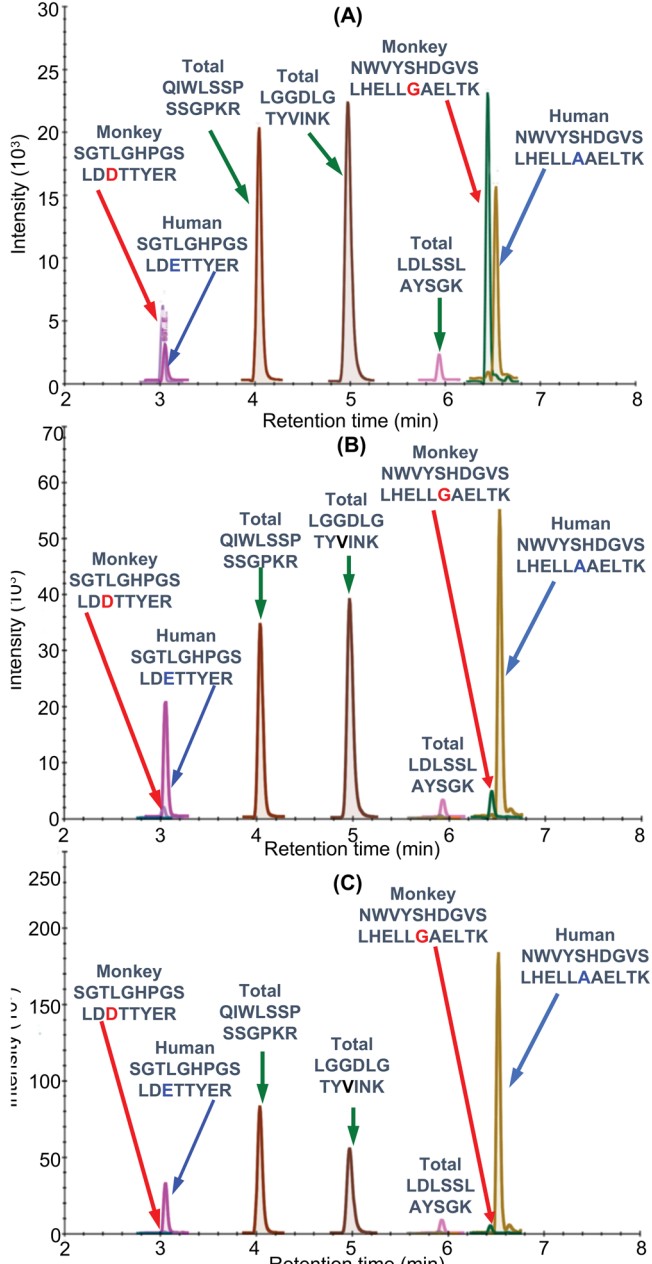

**Fig. 5 UHPLC-MRM/MS analysis of FXN tryptic peptides in monkey hearts after gene therapy.** The same MRM transitions as those shown in the legend to Fig. 3 were used. **A** hFXN-M (1.5 ng/mg tissue) and mFXN-M (3.7 ng/mg tissue) in the right ventricle of monkey # 2 (RV2 B7, 180616) after gene therapy with the lowest dose of 1.00E13 AAV vector GC/kg. **B** hFXN-M (40.1 ng/mg tissue) and mFXN-M (3.4 ng/mg tissue) in the left ventricle of monkey # 3 (LV3 B13, 180701) after gene therapy with the middle dose of 3.00E13 AAV vector GC/kg. **C** hFXN-M (77.0 ng/mg tissue) and mFXN-M (2.3 ng/mg tissue) in the heart septum of monkey # 5 (HS 5, B24, 181271) after gene therapy with the highest dose of AAV vector of 1.00E + 14 GC/kg.

**Non-linear hFXN-M expression with increasing dose of AAV vector.** Rhesus macaque heart tissue samples from the left atrium, right ventricle, left ventricle, and septum were analyzed for hFXN-M after the lowest dose of AAV vector (1.00E13 GC/kg). There were no consistent differences in levels in hFXN-M expression from the four different areas of the heart, they varied from 1.5 to 9.8 ng/mg tissue with a mean of 3.8 ± 1.1 ng/mg tissue

(Fig. 7A, Supplementary Data 5). Similarly, there were no consistent differences in levels in mFXN-M expression from the four different areas of the heart, they varied from 2.4 to 7.9 ng/mg tissue with a mean of 4.1 ± 2.8 ng/mg tissue (Fig. 7A, Supplementary Data 5). Therefore, this dose of vector resulted in levels of hFXN-M that were very similar to the levels of endogenous mFXN-M. It should however be noted that plateau levels were unlikely reached in this short-term 28-day study. A second analysis of hFXN-M levels in the heart was conducted after the middle dose of AAV vector (3.00E13 GC/kg). There were no consistent differences in levels in hFXN-M expression from the four different areas of the heart, they varied from 20.2 to 67.8 ng/mg tissue with a mean of 37.9 ± 17.6 ng/mg tissue (Fig. 7B, Supplementary Data 5). Similarly, there were no consistent differences in levels in mFXN-M expression from the four different areas of the heart, they varied from 2.6 to 7.0 ng/mg tissue with a mean of 4.3 ± 1.3 ng/mg tissue (Fig. 7B, Supplementary Data 5). Therefore, this dose of vector resulted in levels of hFXN-M that were 8.8-fold higher than the levels of endogenous mFXN-M. Finally, an analysis of hFXN-M levels in heart tissues was conducted after the highest dose of AAV vector (1.00E14 GC/kg). There were no consistent differences in levels in hFXN-M expression from the four different areas of the heart; however, there was one tissue sample where there was very little hFXN-M expression (4.0 ng/mg tissue) (Fig. 7C, Supplementary Data 5). The levels varied from 4.0 to 86.3 ng/mg tissue with a mean of 67.7 ± 15.4 ng/mg tissue when the one outlier was excluded (Fig. 7C, Supplementary Data 5). Similarly, there were no consistent differences in levels in mFXN-M expression from the four different areas of the heart, they varied from 0.6 to 4.2 ng/mg tissue with a mean of 2.8 ± 1.2 ng/mg tissue (Fig. 7C, Supplementary Data 5). Therefore, this dose of vector resulted in levels of hFXN-M that were 24.0-fold higher than the levels of endogenous mFXN-M. There appeared to be a reduction in the amount of mFXN-M that was expressed in the heart, which might have been due to the problem of quantifying mFXN-M in the presence of such high levels of hFXN-M. However, the hFXN-M levels expressed in the heart were still 32.2-fold higher than the mFXN-M levels we found in control monkey hearts (2.1 ng/mg tissue, Fig. 4e, Supplementary Data 4).

## Discussion

Despite several relevant reports[30–32] and a nonbinding guidance from the US Food and Drug Administration (FDA) on how to conduct long-term gene therapy studies[33], the pharmacology of gene therapy is still in its infancy. The FDA suggests that long-term plans for monitoring patients who receive gene therapies should be included in study protocols for specific cases described by a decision tree in the guidance[33]. Modeling and pharmacokinetic analysis could significantly aid in the planning for these protocols. The objective of gene therapy is to express DNA or RNA constructs at a desired tissue site in vivo in order to evoke sustained protein expression at levels that can safely achieve a therapeutic effect[31]. To inform dosing decisions, it has been customary to use mRNA levels as a surrogate indicator of tissue protein expression, despite numerous reports that there is a poor relationship between mRNA levels and protein expression in tissues. For example, it was found in a xenograft model system differentially expressed mRNAs correlated significantly better with the levels of expressed protein than non-differentially expressed mRNAs[34]. Another study showed that that protein concentrations correlated with the corresponding mRNA levels by only 20–40%, and that mRNA abundances were poor predictors of protein expression levels[35]. Furthermore, the relationship between mRNA levels and protein expression in T-cells can

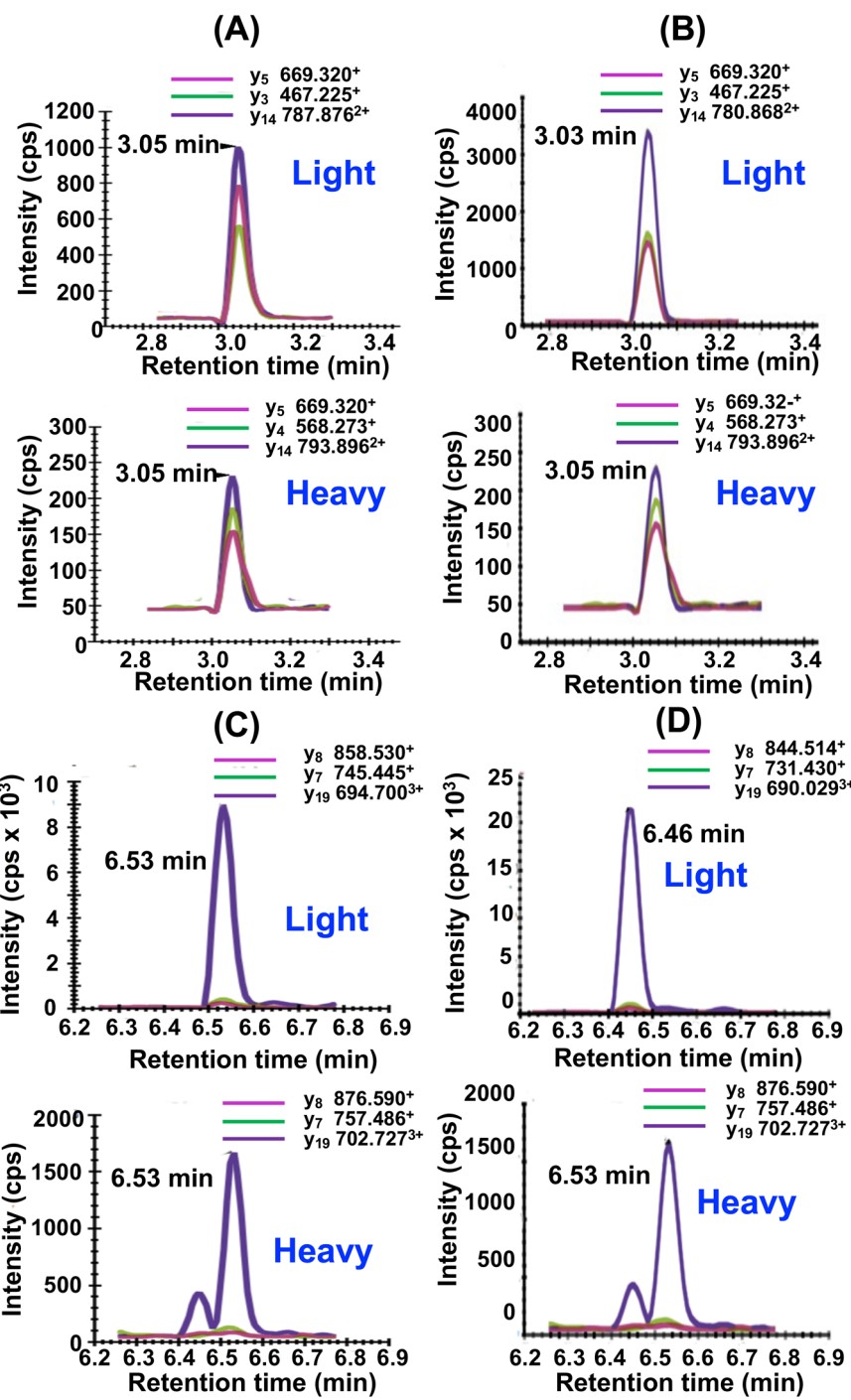

**Fig. 6 Typical UHPLC-MRM/MS chromatograms of tryptic peptides used for quantification of FXN.** There was 1.5 ng/mg tissue of hFXN-M and 3.7 ng/mg tissue of mFXN-M in the right ventricle of monkey # 2 (RV2 B7, 180616) after the lowest vector dose of vector of 1.00E13 GC/kg. **A** SGTLGHPGSLD**E**TTYER tryptic peptide from hFXN-M (MH$_3$$^{3+}$ = 607.287; upper) and heavy SGT**L**GHPGS**L**D**E**TTYER internal standard (MH$_3$$^{3+}$ = 611.300; lower). **B** NWVYSHDGVSLHELL**A**AELTK tryptic peptide from hFXN-M (MH$_4$$^{4+}$ = 596.308; upper) and heavy NWVYSHDGVS**L**HE**LL**A**A**EL**TK internal standard (MH$_4$$^{4+}$ = 602.328; lower). **C** SGTLGHPGSLD**D**TTYER tryptic peptide from mFXN-M (MH$_3$$^{3+}$ = 602.615; upper) and heavy SGT**L**GHPGS**L**D**E**TTYER internal standard (MH$_3$$^{3+}$ = 611.300; lower). **D** NWVYSHDGVSLHELL**G**AELTK tryptic peptide from mFXN-M (MH$_4$$^{4+}$ = 592.804; upper) and heavy NWVYSHDGVS**L**HE**LL**AAE**L**TK internal standard (MH$_4$$^{4+}$ = 602.328; lower). **L** = [$^{13}$C$_6$]-leucine.

be gene class specific and associated with particular amino acid sequence characteristics[36]. Protein levels are also affected by different conditions such as whether tissues are in steady state, undergoing long-term state changes or subjected to acute perturbations[37]. Considering there is a documented poor relationship between differential mRNA and protein expression, there is a need for a reliable and direct method to quantify tissue

protein expression to authoritatively inform gene therapy dose selection decisions.

Several previous studies have addressed the relationship between gene therapy and expressed protein levels. For example, it was reported that a single dose of AAV gene therapy resulted in sustained serum levels of immunotherapeutic proteins[38]. The study reported that the gene therapy resulted in sustained protein

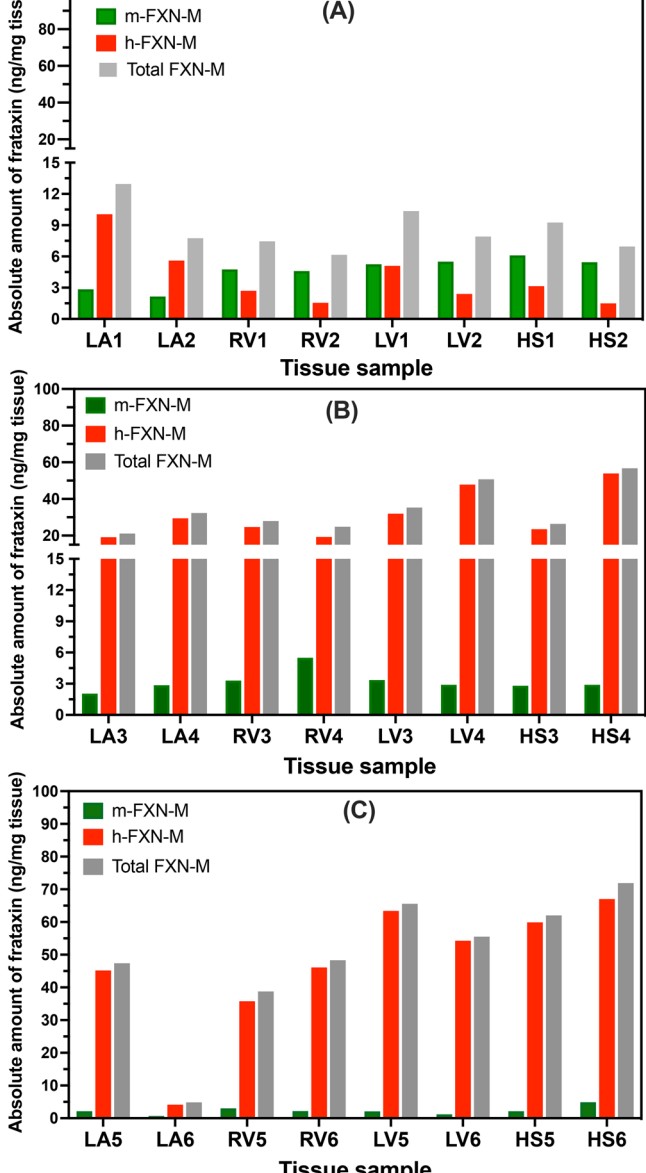

**Fig. 7 Levels of hFXN-M and mFXN-M in Rhesus macaque heart tissues after IV administration with increasing doses of AAV vector. A** Lowest dose of 1.00 + E13 AAV vector genome copies (GC)/kg in two monkeys. **B** Middle dose of 3.00E + 13 GC/kg in two monkeys. **C** Highest dose of 1.00E + 14 GC/kg in two monkeys. Left atrium = LA; Right ventricle = RV; Left ventricle = LV; Heart septum = HS.

expression in the systemic circulation for up to 1 year in mice. Another study addressed the use of AAV gene therapy for systemic protein delivery, where it was suggested that further investigation into AAV expression at the tissue level would be the key to understanding how the system can be perturbed for increased potency[39]. In addition, a tunable switch system to control levels of gene therapy expression has been described, which could be used to optimize protein tissue levels[40]. These studies collectively suggest that gene therapy can result in sustained tissue protein expression. However, there are no previous studies that directly quantify human protein tissue levels after gene therapy in non-human primates. Therefore, the current study is the first to specifically quantify human tissue protein expression after gene therapy, providing insight into why this is

so important because of the consequences of non-linear pharmacokinetics.

Preclinical gene therapy studies in animal models are required before the vectors can be tested in humans. Non-human primates offer significant advantages over rodent and canine models especially in models of rare genetic diseases[41]. Non-human primate studies are essential for the safety assessment of AAV-based gene therapy products prior to human studies[42]. They have also been very useful for assessing the neuropathology associated with gene therapy targeted at the central and peripheral nervous system[43], and for assessing the immune response to helper dependent adenoviral mediated liver gene therapy[44]. Importantly, the expressed proteins from non-human primates are generally very similar in sequence to human proteins, which limits the potential immunogenicity of a foreign protein[42]. In the case of hFXN-M, there are only two amino acid differences when compared with mFXN-M, so the sequences are 98.5% identical (Fig. 1). In contrast, mouse FXN-M has 12 amino acid differences when compared with hFXN-M, so the sequences are only 90.8% identical[22]. Furthermore, mouse FXN-M is expressed as two major proteoforms in the mitochondria and cytosol of cells from mouse heart tissue, whereas hFXN-M is only found as a single proteoform in the mitochondria of cells from human heart tissue[22]. The advantage of using non-human primates for hFXN-M gene therapy studies presents a formidable bioanalytical challenge in quantification of proteins with almost identical sequences. One needs to be able to quantify precisely and distinguish both the endogenous non-human primate protein and the almost identical therapeutic gene product human protein. In the case of hFXN-M and mFXN-M, we were able to develop a novel method to distinguish and quantify the two proteins using stable isotope dilution IP-LC-MRM/MS (Fig. 2), offering unprecedented insight in transgene product quantification using a species relevant protein for pharmacology and safety evaluation.

hFXN-M concentrations were determined from a standard curve prepared by adding increasing amounts of hFXN-M to the same amount of SILAC hFXN-M that was added to the tissue samples. Therefore, SILAC hFXN-M in effect, acts as the loading control because the tryptic peptides are identical to those derived from hFXN-M, differing only in mass. The ratio between hFXN-M and mFXN-M expressed in mouse heart tissue to the SILAC hFXN-M internal standard is established at this time. Therefore, corrections do not have to be made for any losses during the subsequent isolation and analysis procedure. The FXN-M amounts were calculated from the ratio of hFXN-M and mFXN-M tryptic peptides to the corresponding SILAC tryptic peptides (Fig. 4a–d and Fig. 6A–D) using the regression lines of relevant standard curves (Fig. 2B, C). The standard curves were prepared at the time of the analysis using the same isolation procedure as used for the tissue samples. The precision and accuracy of these measurements are provided in Table 2.

The SILAC-based UHPLC-MS method is designed to quantify proteins in biofluid and tissue samples by establishing the ratio between the SILAC internal standard and endogenous analytes at the time the biofluid or tissue sample is thawed[14]. Isolation of the various cell types would result in significant FXN-M protein loss and would not be reflective of the FXN-M concentration at the time of sample collection. We found highly consistent levels of hFXN-M (5.1 ± 1.7 ng/mg tissue) in control human heart tissues as well as highly consistent levels of mFXN-M (2.1 ± 0.4 ng/mg tissue) in control monkey heart (Fig. 4e, Supplementary Data 4). hFXN-M and mFXN-M are only found in the mitochondria because they require MPP (which is only found in the mitochondria) to process the full-length FXN into these bioactive forms[5,6]. If there were selective loss of certain cells in the heart tissues that were analyzed, or if there were a different distribution of cells compared with the

intact heart, this would have been reflected in aberrant levels of FXN-M. The consistency of FXN-M levels in the different human and monkey heart tissues argues against this possibility. Furthermore, there is no evidence that full-length FXNs are differentially targeted to mitochondria in different cell types. Therefore, the ratio between hFXN-M and mFXN-M unequivocally establishes whether hFXN-M is being overexpressed compared with the endogenous mFXN-M expressed in the mitochondria of target heart tissue cells. There could be several reasons why there was decreased expression of hFXN-M in sample LA6 (Fig. 7C, Supplementary Data 5). However, this was not due to selective cell loss in the tissue sample, which would have resulted in aberrant levels of mFXN-M.

Previous gene therapy studies conducted in mouse models suggested that toxicity occurred through over-expression of hFXN-M protein and that there was also a non-linear dose-response[21]. Paradoxically, the toxicity appeared to be associated with a decrease in iron-sulfur cluster complexes[45], although the biological activity of FXN-M involves (in part) the assembly of these complexes[46]. The mean level of hFXN-M in five control (non-FRDA) human hearts was found to be $5.1 \pm 1.7$ ng/mg tissue in the present study (Fig. 4e, Supplementary Data 4). The mean level of mFXN-M in control monkey heart was $2.1 \pm 0.4$ ng/mg tissue (Fig. 4e, Supplementary Data 4). The lowest dose of AAV vector of 1.00E13 GC/kg resulted in the expression of hFXN-M at a mean level of $4.1 \pm 2.8$ ng/mg tissue (Fig. 7A, Supplementary Data 5) which is close to the endogenous level of hFXN-M in human heart (Fig. 4e, Supplementary Data 4) and mFXN-M in monkey heart (Fig. 4e, Supplementary Data 4). In contrast, a three-fold increase in dose of the AAV vector (3.00E13 GC/kg) resulted in an almost 10-fold increase in the level of hFXN-M protein ($37.9 \pm 17.6$ ng/mg protein) in heart tissues compared with endogenous mFXN-M ($4.1 \pm 2.8$ ng/mg protein) in the same tissues (Fig. 7B, Supplementary Data 5). Furthermore, heart tissue hFXN-M levels ($67.7 \pm 15.4$ ng/mg protein) after a ten-fold dose increase in vector (1.00E14 GC/kg) were 20-fold higher than the endogenous levels of mFXN-M ($2.8 \pm 1.2$ ng/mg protein) in the monkey heart (Fig. 7C, Supplementary Data 5). These potentially toxic concentrations with modest increases in the AAV vector dose illustrate the need for careful pharmacokinetic studies prior to using vector in clinical studies to treat patients. Furthermore, even with limited sample sizes and the short 28-day study duration, it was clearly evident that the dose administered did not predict the observed hFXN-M protein levels. At the lowest dose of vector, no toxicity would be predicted from the observed hFXN-M levels (Fig. 7A, Supplementary Data 5), although longer timepoints and increased sample sizes would be needed to fully assess the possible toxicity of such hFXN-M levels.

On the positive side, our study, which has rigorously quantified both the human and monkey protein in a gene therapy study, has shown that the lowest dose of vector resulted in the expression of hFXN levels in monkey hearts at similar levels to those found in control (non-FRDA) human and monkey hearts. It is plausible that these levels of hFXN-M in human hearts would prevent the cardiomyopathy associated with FRDA. Furthermore, the highly sensitive and specific stable isotope dilution IP-UHPLC-MRM/MS that was developed in the current study will allow hFXN-M to be quantified in only 1 mg human heart biopsy tissue taken before and after administration of the AAV vector to FRDA patients. Such studies are currently under way. Finally, insights from the present study can be applied to all gene therapy studies, which will allow essential pharmacokinetic parameters to be determined in non-clinical models as well as in patients undergoing gene therapy, and to objectively compare expression levels achieved across different studies. We anticipate that this will simplify and expedite the selection of the most appropriate gene therapy dose

to optimize efficacy and avoid the toxicity that can occur with over-expression of bioactive proteins.

## Methods

**Materials and reagents**. Protein G Dynabeads for immunoprecipitation, RIPA lysis buffer with EDTA, formic acid Optima LC/MS Grade, and CaptureSelect™ AAV9 resin were from Thermo-Fisher Scientific (Waltham, MA). Phosphate-buffered saline (PBS) ammonium bicarbonate ($NH_4HCO_3$), dimethyl pimelimidate dihydrochloride (DMP), ethanolamine, triethylamine (TEA), 0.2 M cross-linking buffer, pH 8.0, dithiothreitol (DTT), acetic acid (glacial), Tween-20, EDTA-free Easypack protease inhibitor cocktail tablets, bovine serum albumin (BSA), and sodium dodecyl sulfate (SDS) were from MilliporeSigma (Billerica, MA). Trypsin endoproteinase sequencing grade was from Promega (Madison, WI), acetonitrile (ACN), LC/MS grade water and Optima LC/MS grade, were from Fisher Scientific (Pittsburg PA). Mouse anti-frataxin monoclonal antibody (mAb) Ab113691 raised against the N-terminal sequence of hFXN was from Abcam (Walton, MA). hFXN-M protein and SILAC-hFXN-M protein internal standard were prepared as previously described[47]. De-identified fresh frozen control (non-FRDA) heart tissue was supplied by BioIVT (Hicksville, NY) from consented donors and the relevant IRB approval was obtained. Subjects had typically died from heart failure.

**AAV vector**. The Penn Vector Core produced and titrated AAV vectors for the study as previously described[48]. In brief, HEK293 cells were triple-transfected, and the culture supernatant was harvested, concentrated, and purified with an iodixanol gradient. The vector was produced by triple transfection of adherent HEK293 cells and purified from the supernatant by affinity chromatography using a POROS™ CaptureSelect™ AAV9 resin, followed by anion exchange chromatography. Limulus amebocyte lysate and quantitative polymerase chain reaction (qPCR) tests for endotoxin and mycoplasma, respectively, were negative. Vector titer by TaqMan PCR was $6.05 \times 10^{13}$ genome copies (GC)/mL. The purity of capsid proteins was 95.34%, as determined by sodium dodecyl sulfate-polyacrylamide gel electrophoresis analysis. Purified vectors were titrated with droplet digital PCR using primers targeting the rabbit beta-globin polyA sequence as previously described[49].

**Animal dosing**. The use of Rhesus macaque (Macaca mulatta) monkeys in the study was approved by IUCAC of the University of Pennsylvania Approval #806620. The macaques were procured from Orient BioResource Center Inc. via PreLabs. Animals were 3.6–4.5 years old and weighed between 4.6–5.5 Kgs at study initiation. The animals were housed in the AAALAC International-accredited Nonhuman Primate Research Program facility at the University of Pennsylvania in stainless steel squeeze-back cages as groups. Animals received varied enrichments such as food treats, visual and auditory stimuli, manipulatives, and social interactions. We complied with all relevant ethical regulations for animal testing of Rhesus macaques in the age range of 3.6–4.5 years. The monkeys received a single intravenous injection (Saphenous vein) of the vector at 1.00E13, 3.00E13, and 1.00E14 GC/Kg.

**Tissue samples**. At study day-28, animals were euthanized, and necropsies were performed. Hearts were removed from the animals and samples from ventricles, aorta, and septum were collected and immediately frozen to $-80\,°C$ and stored at this temperature until analyzed.

**Preparation of DMP crosslinked Dynabeads**. A Dynabead suspension (5 mg, 150 μL) was transferred to a Sarstedt 2.0 mL low bind (LB) microtube (Sarstedt, AG Numbrecht, Germany) and washed three times using 500 μL of bead washing buffer (PBS with 0.02% Tween-20). Mouse anti-FXN mAb Ab113691 (40 mg, 80 μL) was diluted using PBS to a final volume of 500 μL in a Sarstedt 2.0 mL LB microtube. The beads were incubated with the mouse mAb at 4 °C overnight on a Mini LabRoller rotator (Labnet International, Edison, NJ). The beads were swirled so that they were thoroughly suspended, the mAb solution removed, and the beads washed twice with 1 mL of the 0.2 M TEA cross-linking buffer. A solution of DMP was prepared by dissolving 13 mg of DMP in 2 mL cross-linking buffer (freshly made every time). The beads were incubated with the 2 mL DMP solution at room temperature (RT) in a Sarstedt 2.0 mL LB microtube for on the rotator. After 1 h, the DMP solution was removed and the beads washed with 1 mL of quenching buffer (0.1 M ethanolamine, pH 8.0; 301 μL in 50 mL water). This was followed by incubation of the beads with 1 mL of quenching buffer at RT on the rotator for 1 h, removal of the quenching buffer and washing the beads twice with 1 mL of bead washing buffer. DMP cross-linked beads were used immediately.

**IP of heart tissue samples**. DMP cross-linked beads were gently well-mixed in 1 mL of bead wash buffer then pipetted in 100 μL aliquots (0.5 mg beads/sample) into Sarstedt 2.0 mL LB microtubes. Monkey heart tissue samples (20 mg to 200 mg) were weighed and transferred to a Sarstedt 2.0 mL LB microtube. The tube was washed and dried, accurately re-weighed, and the amount of heart tissue recorded. RIPA lysis buffer (500 μL) containing the protease inhibitor cocktail was added to heart tissue followed by approximately, 30–50 stainless steel beads (0.9–2.0 mm). Homogenization was conducted using the Bullet Blender Gold homogenizer (Next Advance, Troy, NY) at a speed of 10 for 5 min at 4 °C. Samples were then lysed completely by additional probe sonication on ice with 30 pulses at power 4 using a sonic dismembranator (Fisher Scientific, Pittsburgh, PA). The SILAC-hFXN-M internal standard solution (20 μL of 2 μg/mL, 40 ng) was then added to each sample and appropriate amounts of hFXN-M standards 4 ng to 200 ng (4, 10, 15, 20, 30, 40, 80, 100, 150, 200 ng) in 5% BSA solution for preparation of the standard curve[13]. Supernatants were removed from the DMP cross-linked beads and tissue and BSA standard samples added into the Sarstedt 2.0 mL LB microtubes containing the DMP cross-linked beads. Samples were then incubated with the beads at 4 °C overnight on the rotator.

**Elution of hFXN-M and mFXN-M from the beads**. Supernatants with unbound proteins were removed and the beads resuspended in bead wash buffer (1 mL). The beads were washed twice with bead wash buffer (1 mL) then transferred to clean 1.5 mL microcentrifuge tubes (ThermoFisher Scientific, Waltham, MA). After washing the beads with PBS (1 mL), hFXN-M and mFXN-M were eluted with 200 μL of elution buffer (100 mM acetic acid in 10% ACN) by shaking the beads in a thermal mixer (ThermoFisher Scientific, Waltham, MA) at 1000 rpm and RT for 1 h. Supernatants were then transferred to Sarstedt 2.0 mL LB microtubes and dried under a nitrogen flow using an N-Evap concentrator (Organomation, Berlin, MA).

**Trypsin digestion of eluted hFXN-M and mFXN-M**. Dried eluates from the Dynabeads were dissolved in 50 mM NH₄HCO₃ solution (50 μL) containing trypsin (100 ng). Samples were Incubated at 37 °C for 18 h then centrifuged at 15,000 g for 2 min on a bench-top centrifuge (Eppendorf North America, Enfield,

CT) at RT. Supernatants were transferred to Waters 150 μL insert tubes with pre-installed plastic springs (Waters, Milford, MA), which were then placed into Waters LC-MS certified PTFE/silica 2 mL screw-top injection vials ready for analysis by UHPLC-MRM/MS.

**UHPLC-MRM/MS analysis**. Solutions (2 μL) containing the tryptic peptides were injected in triplicate on the UHPLC-MRM/MS system. This was equivalent to the injection of a trypsin digest from heart tissue spiked with 1.6 ng (105 fmol) of SILAC-hFXN-M protein internal standard. An Agilent 1290 Infinity UHPLC system equipped with a Zorbax Rapid Resolution High Definition (2.1 × 50 mm, 1.8 μm particle size) UHPLC column was used. It was coupled on-line to an Agilent 6495 C triple quadrupole mass spectrometer. Mobile solvent A was 0.1% formic acid in water and solvent B was 0.1% formic acid in ACN. The UHPLC column was maintained at 35 °C with a flow rate of 0.4 mL/min. The tryptic peptides were separated with the following linear gradient: 5% B at 0-min, 5% B at 1-min, 24% B at 2.75-min, 36% B at 3.50-min, 95% B at 5.0-min, 95% B at 6.50-min, 5% B at 7.0-min, 5% B at 8.5-min ready for the next injection. Mass spectrometer operating conditions were maintained as follows: nitrogen gas flow 13 L/min, gas temperature 230 °C, nebulizer gas 40 psi, sheath gas temperature 300 °C, sheath gas flow 10 L/min, capillary voltage 4500 V, and nozzle voltage 500 V. The MRM/MS transitions shown in Table 1 were used and care was taken to ensure that the retention times (ret time) were within 0.1 min of the times shown in Table 1.

**Data analysis**. Peptide quantification was performed using Skyline (MacCoss Laboratory, University of Washington, Seattle, WA)[50]. The peak area ratio of each MRM transition for each unlabeled/light (L) peptide to labeled/heavy (H) peptide was calculated by the Skyline software and used for absolute quantification. The peptide ratios were calculated from the sum of L/H ratios of the MRM transitions of the $y_{14}^{2+}$-, $y_5^+$-, and $y_3^+$-ions specific for SGTLGHPGSLD**E**TTYER the N-terminal tryptic peptide specific for hFXN-M (blue signifies the amino acid specific to hFXN) and $y_{14}^{2+}$-, $y_5^+$-, and $y_4^+$-ions for the heavy SGT**L**GHPGS**L**D**E**TTYER internal standard as well as and $y_{19}^{3+}$-, $y_8^+$-, and $y_7^+$-ions from NWVYSHDGVSLHELL**A**AELTK tryptic peptide specific for hFXN-M and $y_{19}^{3+}$-, $y_8^+$-, and $y_7^+$-ions from the heavy NWVYSHDGVS**L**HE**LLA**AE**L**TK internal standard (lower). The peptide ratios were then used to calculate the mean amount of hFXN-M from the relevant standard curves (Fig. 2B, C). Similarly, the peptide ratios were calculated from the sum of L/H ratios of the MRM transitions of the $y_{14}^{2+}$-, $y_5^+$-, and $y_3^+$-ions of the SGTLGHPGSLD**D**TTYER N-terminal tryptic peptide from mFXN-M (red signifies the amino acid specific to mFXN) and $y_{14}^{2+}$-, $y_5^+$-, and $y_4^+$-ions from the heavy SGT**L**GHPGS**L**D**E**TTYER internal standard as well as the $y_{19}^{3+}$-, $y_8^+$-, and $y_7^+$-ions from NWVYSHDGVSLHELL**G**AELTK tryptic peptide from mFXN-M and $y_{19}^{3+}$-, $y_8^+$-, and $y_7^+$-ions from the heavy NWVYSHDGVS**L**HE**LLA**AE**L**TK internal standard. The peptide ratios were then used to calculate the mean amount of mFXN-M from the relevant standard curves (Fig. 2B, C). Standard curve for this peptide constructed using triplicate injections of known amounts of hFXN-M in the range 4 ng to 200 ng (4, 10, 15, 20, 30, 40, 80, 100, 150, 200 ng) with 40 ng of SILAC-hFXN-M as internal standard extracted by IP from BSA.

**Statistics and reproducibility**. The slopes, intercept, coefficient of determination ($r^2$), the back-calculated accuracy, SDs and the precision (% CV) of the standard curves were determined from five replicates of SGTLGHPGSLD**D**TTYER (Fig, 2B, Table 2A,

Supplementary Date 2) and NWVYSHDGVSLHELL**G**AELTK (Fig. 2C, Table 2B, Supplementary Data 3). Three MRM/MS transitions for three other tryptic peptides common to both FXN-M and mFXN-M (QIWLSSPSSGPKR, LGGDLGTYVINK, and LDLSSLAYSGK) were monitored to provide additional confirmation that FXN-M was present. SDs were determined with GraphPad Prism 10 for Mac OS version 10.0.3. Heart tissue samples were obtained from two animals at each dose. Each heart tissue sample was analyzed in duplicate. Three replicate injections of the two duplicates were made on the LC-MS system and the mean FXN-M value from each duplicate was then used to calculate the tissue level in ng/mg protein.

**Reporting summary.** Further information on research design is available in the Nature Portfolio Reporting Summary linked to this article.

## Data availability

The data that support the findings of this study are available within the paper. Numerical source data underlying graphs and plots in the manuscript can be found in supplementary data files 1–5. Any additional information not included in the paper is available upon request from Dr. Ian. A. Blair or Dr. James M. Wilson.

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

## Acknowledgements

We thank Dr. Kevin Horgan for insightful discussions, the Finneran and Hamilton families for financial support, and Agilent Technologies Inc. for the loan of an Agilent 6495 C triple quadruple mass spectrometer. We also acknowledge the financial support of the National Institute of Neurological Disorders and Stroke U01NS114143, National Institute for Environmental Health Sciences P30ES013508, and FA212, LLC (Rye, New York, NY).

## Author contributions

I.A.B., J.M.W. J.J.H., and C.H. designed the study and experiments, C.H. and G.R.C. conducted the animal experiments, T.R. and C.M. conducted the mass spectrometry analyses, I.A.B. and C.M. performed the data analysis. I.A.B., J.M.W. and J.J.H. wrote the manuscript. All authors discussed the results, read the manuscript, and made revisions.

## Competing interests

The authors declare the following competing interests: J.M.W is a paid advisor to and holds equity in iECURE, Scout Bio, Passage Bio, and the Center for Breakthrough Medicines (CBM). He also holds equity in the former G2 Bio asset companies. He has sponsored research agreements with Amicus Therapeutics, CBM, Elaaj Bio, FA212, former G2 Bio asset companies, iECURE, Passage Bio, and Scout Bio, which are licensees of Penn technology. C.J.H. holds equity in Scout Bio and a former G2 Bio asset company. J.M.W., C.J.H, and J.J.H, are inventors on patents that have been licensed to various biopharmaceutical companies and for which they may receive payments. I.A.B. has sponsored research agreements with Lexeo Therapeutics and Design Therapeutics. All other authors declare no competing interest.
