## [Peer Review File · Communications Biology]

Reviewers' comments:

Reviewer #1 (Remarks to the Author):

Summary:

The manuscript by authors Rojsajjakul et al, titled "Quantification of human mature frataxin protein expression in nonhuman primate hearts after gene therapy" presents their work developing an assay to quantify efficacy of frataxin gene-replacement therapy in the hearts of a non-human primate (Rhesus macaques) using immunopurified and SILAC-FXN controlled mass spec analyses. The motivation for this study is to validate an approach to overcome challenges of quantifying scarce proteins in from a complex tissue (the heart) from a higher-level vertebrate species and distinguish abundance of the endogenous from the exogenous proteins where the amino acid consensus is 98%. They provide results showing the feasibility of this technical approach in distinguishing between endogenous and exogenous frataxin protein over 3 dose-levels of AAV hu68.CB7 vector. Furthermore, they provide evidence for a possible non-linear dose-response expression at the protein level.

Overall Impression:

This manuscript describes a well-executed and rational approach to making precise measurements of the efficacy of virally mediated gene replacement therapy in the desired tissue type, at the protein level. It represents a useful technological methods paper that will be of interest to scientists and pharmaceutical companies developing such therapies for a wide range of conditions. I applaud the use of mass spec and SILAC standards to quantitatively distinguish between the frataxin of the host and the AAV vector. I also appreciate the importance of performing such experiments in a species closer to human than the usual mouse models often employed. The results suggesting a non-linear dose-response to protein expression is interesting and potentially important. Nevertheless, the manuscript has limitations that significantly diminish enthusiasm. These concerns are outlined in the Specific Comments section.

Specific Comments:

1. The first concern is the limited scope and preliminary nature of the experimentation. Only 2 animals at each dose were used and analyzed at a single time point (28-days after AAV injection), which as the authors acknowledge, not likely long enough to reach plateau expression level. As such, the work is more of a "proof of feasibility" than a thorough investigation with firm conclusions. It appears not much more than a reapplication of the methods this group (PMID: 36800320) to the heart assessing viral mediated gene expression, diminishing its novelty.
2. The use of SILAC frataxin standardization is good but there are no other controls for tissue/cell/organelle loading. Accordingly, the precision of the nanogram/mg are less meaningful. The heart is a complex organ wherein only ~50% of the cells are actually myocytes. Furthermore, the distribution of myocytes and non-myocytes (endothelial cells, vascular smooth muscle cells, cardiac fibroblasts, resident macrophages) is not uniform and subject to sampling error. The mitochondrial abundance may also differ among the chambers as well (particularly in the more energetically demanding ventricular compared to that of the atrium). So, I cannot feel confident that the real concentration of frataxin in the cells of interest (myocytes) among the tissues sampled is accurate. This is particularly exemplified in the sample LA6, figure 7C. Potential solutions to this could be affinity purification or FACS of myocytes prior to protein isolation. I would also advise myocyte-specific and mitochondrial specific marker proteins to assess loading conditions more accurately. Such controls would likely give more meaningful frataxin concentrations.
3. An important paper (PMID: 33209958) is cited that first demonstrated that over-expression of frataxin was toxic/pathogenic in heart of mouse models. The authors of the present manuscript point out shortcomings of that paper (in ability to distinguish between mouse and human frataxin abundance) as justification/rationale for their study. While there are shortcomings to that published paper, it was able to specifically determine the relative abundance of exogenous frataxin by either HA-tag or by using the conditional knockout Mck mouse. Moreover, this paper provided evidence of a similar non-linear dose response with AAV-FXM. Accordingly, the statements in both the introduction (page4, lines 89-93) and discussion (page 14, lines 332-333) are misleading.

4. The important finding of a possible non-linear dose response of protein expression merits further investigation. Increased animal numbers, longer duration, serial sampling (myocardial biopsy rather than single time point animal sacrificing), independent assay for viral DNA abundance and persistence. Consideration of a weaker or cardiomyocyte-specific promoter might be helpful in determining the mechanism. Toxicity will be very important, but admittedly, this will likely take much longer. Nevertheless, I would advise toxicity studies to focus on lower-level doses (2-10-fold increase in frataxin) to truly determine the lowest dose that might be toxic.

Specific & technical points:

- a) Calculation of the fold-increase in frataxin with differing AAV dosage might be done differently. I'm not sure but it seems you are calculating the amount increase over the endogenous frataxin. I would have used the amount of frataxin at the lowest dose as the baseline (or assign it as 1) and do fold change for other doses over that. In this case, my calculation is 9.5-fold at the mid-level dose and 16.75-fold at the highest dose. But as I stated in comment 2 above, these calculations will be subject to the precision of actual protein relative to a loading standard.
- b) On page 14, line 348 there is a statement "It is evident that mRNA data did not predict the observed protein levels". What data do you speak of? Is it from these samples? If so, it should be shown.
- c) On page 15, line 349, the statement "no toxicity was observed" should be supported by some data.
- d) In figure 1, what does the brown arrow signify?
- e) In figure 5, panels A, B, and C, seem to come from different chambers. Why such inconsistency?
- f) Which parts of the heart do the samples in figure 3 come from?
- g) In the abstract, I may have missed it but has the pathogenesis of disease in FRDA been exclusively assigned to mature frataxin? If not, it might be best to just say frataxin without such a modifier.
- h) In the abstract "up to 1400 GAA triplet repeats" may not be entirely accurate. No upper limit has been definitively assigned to the trinucleotide repeat expansion. It might be best to say something like "from 70 to >1000 repeats".

Reviewer #2 (Remarks to the Author):

In this work, Rojsajakul and cols develop a method to specifically quantify the mature frataxin protein after gene therapy studies. This work covers one of the remain questions we still have in the gene therapy field regarding the amount of protein is produced from the AAV vector after treatment and how this "exogenous protein" is distinguish from the endogenous one.

Methodologically this work is brilliant, results are convincing and conclusions very well argued. However, I do still have a few points that should be addressed before accepting it:

- As you shown in the results section, once low and medium dose of the vector was injected, the amount of monkey FXN (mFXN) was not affected. However, it seems there is a reduction of mFXN after the injection of the high dose of the vector was done. This could be a side effect due to the dose used or maybe a limitation in the method to quantify FXN when this protein is highly overexpressed. I wonder whether you have an explanation for this and if so, please explain it in the manuscript.

- In the Belbella study (2020) they showed toxicity in the heart when 20-fold of FXN overexpression was reached after AAV treatment. They showed this toxicity 12-months after the treatment. In your study you do not detect any abnormality in hearts when the high dose was used, even though the amount of FXN detected was 24-fold higher. You claimed that this is due the length of your study (4 weeks). However, in Belbella study they do not detect anatomical anomalies in the heart either, but several disruptions in the heart at histological levels were detected as well as mitochondria disruptions. I wonder whether you have analysed the tissue collected from your study at histological levels. If you still have some left over cardiac tissue, this could help to detect some abnormalities even though your study was shorter in time.

- Most of the gene-therapy preclinical studies are done first in mice before human no primates, do you think this method of quantification is extrapolated to other species?

- Reference 15 regarding gene-therapy drugs already approved is from 2021, however a new gene therapy treatment was approved by 2022. Please, could you find a more recent reference for this?

- There are a few typos throughout the manuscript: hFXN-M seems to be repeated, (page 4, 2nd paragraph, line 4); "tin" (page 12, line 7); "in" (page 14, lane 4)

Response to Reviewers

Reviewer 1

Summary Comments:

The manuscript by authors Rojsajakul et al, titled “Quantification of human mature frataxin protein expression in nonhuman primate hearts after gene therapy” presents their work developing an assay to quantify efficacy of frataxin gene-replacement therapy in the hearts of a non-human primate (Rhesus macaques) using immunopurified and SILAC-FXN controlled mass spec analyses. The motivation for this study is to validate an approach to overcome challenges of quantifying scarce proteins in from a complex tissue (the heart) from a higher-level vertebrate species and distinguish abundance of the endogenous from the exogenous proteins where the amino acid consensus is 98%. They provide results showing the feasibility of this technical approach in distinguishing between endogenous and exogenous frataxin protein over 3 dose-levels of AAV hu68.CB7 vector. Furthermore, they provide evidence for a possible non-linear dose-response expression at the protein level.

Author’s Response to Summary Comments: This is an excellent summary of our manuscript.

Overall Impression Comments:

This manuscript describes a well-executed and rational approach to making precise measurements of the efficacy of virally mediated gene replacement therapy in the desired tissue type, at the protein level. It represents a useful technological methods paper that will be of interest to scientists and pharmaceutical companies developing such therapies for a wide range of conditions. I applaud the use of mass spec and SILAC standards to quantitatively distinguish between the frataxin of the host and the AAV vector. I also appreciate the importance of performing such experiments in a species closer to human than the usual mouse models often employed. The results suggesting a non-linear dose-response to protein expression is interesting and potentially important. Nevertheless, the manuscript has limitations that significantly diminish enthusiasm. These concerns are outlined in the Specific Comments section.

Response to Overall Impression Comments: We really appreciate the reviewer’s positive comments and fully understand the concerns. We hope that we have managed to adequately assuage these concerns in our responses and revisions to the manuscript.

Specific Comments

1. The first concern is the limited scope and preliminary nature of the experimentation. Only 2 animals at each dose were used and analyzed at a single time point (28-days after AAV injection), which as the authors acknowledge, not likely long enough to reach plateau expression level. As such, the work is more of a “proof of feasibility” than a thorough investigation with firm conclusions. It appears not much more than a reapplication of the methods this group (PMID: 36800320) to the heart assessing viral mediated gene expression, diminishing its novelty.

Answer to Concern 1. Our paper describes (for the first time) the absolute quantification of a human protein expressed in the hearts of nonhuman primates together with the absolute quantification of the endogenous protein. We believe that this makes the study novel, particularly considering that the human and monkey proteins are 98.5 % identical. A pre-print of the article already has had 227 reads, which makes it the most highly read Nature Portfolio pre-print of the

cell therapy, gene therapy, protein mass spectrometry, and peptide mass spectrometry manuscripts that have been under review since June 1, 2023 (Blair et al. *Research Square* 2023 Jun 29:rs.3.rs-3121549; submitted June 29, 2023). This is highly suggestive that other researchers also consider that our study is novel. Our earlier published study (PMID: 36800320) simply quantified FXN-M in human blood and did not face the challenge of distinguishing and quantifying both human and nonhuman primate forms of the proteins in heart tissue.

Furthermore, it is not particularly meaningful to simply show that a gene is expressed in target tissue as this provides no information on whether the biologically active protein is also expressed or whether it is present at therapeutic concentrations. The protein level is dependent on both the rate of expression and the rate of degradation. This means that mRNA levels have little utility for assessing the dose of vector required to express therapeutic (rather than toxic) levels of a therapeutic protein. In contrast, the method that we have described makes it possible to accurately and precisely quantify the amount of protein expressed in target tissue. The editor did not require us to increase the sample size. However, the text has been modified as follows:

Page 16 Line 13

“Furthermore, even with limited sample sizes and the short 28-day study duration, it was clearly evident that the dose administered did not predict the observed hFXN-M protein levels.”

2. The use of SILAC frataxin standardization is good but there are no other controls for tissue/cell/organelle loading. Accordingly, the precision of the nanogram/mg are less meaningful. The heart is a complex organ wherein only ~50% of the cells are actually myocytes. Furthermore, the distribution of myocytes and non-myocytes (endothelial cells, vascular smooth muscle cells, cardiac fibroblasts, resident macrophages) is not uniform and subject to sampling error. The mitochondrial abundance may also differ among the chambers as well (particularly in the more energetically demanding ventricular compared to that of the atrium). So, I cannot feel confident that the real concentration of frataxin in the cells of interest (myocytes) among the tissues sampled is accurate. This is particularly exemplified in the sample LA6, figure 7C. Potential solutions to this could be affinity purification or FACS of myocytes prior to protein isolation. I would also advise myocyte-specific and mitochondrial specific marker proteins to assess loading conditions more accurately. Such controls would likely give more meaningful frataxin concentrations.

Answer to Concern 2: The text has been modified as follows:

Page 15 Line 2

“The SILAC-based UHPLC-MS method is designed to quantify proteins in biofluid and tissue samples by establishing the ratio between the endogenous analyte and the SILAC internal standard at the time the biofluid or tissue sample is thawed.¹⁴ Isolation of the various cell types would result in significant FXN-M protein loss and would not be reflective of the FXN-M concentration at the time of sample collection. We found highly consistent levels of hFXN-M (5.1 ± 1.7 ng/mg tissue) in control human heart tissues as well as highly consistent levels of mFXN-M (2.1 ± 0.4 ng/mg tissue) in control monkey heart (Fig. 4E). hFXN-M and mFXN-M are only found in the mitochondria because they require MPP (which is only found in the mitochondria) to process the full-length FXN into these bioactive forms.^{5,6} If there were selective loss of certain cells in the heart tissues that were analyzed, or if there were a different distribution of cells compared with the intact heart, this would have been reflected in aberrant levels of FXN-M. The consistency of FXN-M levels in the different human and monkey heart tissues argues against this possibility. Furthermore, there is no evidence that full length FXNs are differentially targeted to mitochondria

in different cell types. Therefore, the ratio between hFXN-M expressed in monkey hearts and mFXN-M unequivocally establishes whether hFXN-M is being overexpressed compared with the endogenous mFXN-M expressed in the mitochondria of target heart tissue cells. There could be several reasons why there was decreased expression of hFXN-M in sample LA6 (Fig. 7C). However, this was not due to selective cell loss in the tissue sample as this would have resulted in aberrant levels of mFXN-M.”

3. An important paper (PMID: 33209958) is cited that first demonstrated that over-expression of frataxin was toxic/pathogenic in heart of mouse models. The authors of the present manuscript point out shortcomings of that paper (in ability to distinguish between mouse and human frataxin abundance) as justification/rationale for their study. While there are shortcomings to that published paper, it was able to specifically determine the relative abundance of exogenous frataxin by either HA-tag or by using the conditional knockout Mck mouse. Moreover, this paper provided evidence of a similar non-linear dose response with AAV-FXM. Accordingly, the statements in both the introduction (page4, lines 89-93) and discussion (page 14, lines 332-333) are misleading.

Answer to Concern 3: We agree, the incorrect statements have been removed and the text has been modified as follows:

Page 4 Line 22

“It has been reported that hFXN-M cardiac overexpression up to 9-fold the normal endogenous mouse FXN-M proteoform levels in mice was safe, but significant toxicity to the heart at levels above 20-fold was found²¹. However, the methodology that was used did not establish whether the hFXN-M was truncated in mouse heart, or whether it was present in the cytosol as well as the mitochondria, like mouse FXN-M²². Therefore, we reasoned that further studies were required to firmly establish the relationship between the dose of hFXN-M vector and the expression of intact hFXN-M protein in the mitochondria of target tissue. To address this important issue, we used rhesus macaque monkeys as a nonhuman primate model because hFXN-M and monkey FXN-M (mFXN-M) are 98.5 % identical and both are formed in mitochondria from full length FXN protein in a similar manner (Fig. 1).

Page 15 line 22

“Previous gene therapy studies conducted in mouse models suggested that toxicity occurred through over-expression of hFXN-M protein and that there was also a non-linear dose-response²¹. Paradoxically, the toxicity appeared to be associated with a decrease in iron-sulfur cluster complexes⁴⁴, although the biological activity of FXN-M involves (in part) the assembly of these complexes⁴⁵.

Comment 4. The important finding of a possible non-linear dose response of protein expression merits further investigation. Increased animal numbers, longer duration, serial sampling (myocardial biopsy rather than single time point animal sacrificing), independent assay for viral DNA abundance and persistence. Consideration of a weaker or cardiomyocyte-specific promoter might be helpful in determining the mechanism. Toxicity will be very important, but admittedly, this will likely take much longer. Nevertheless, I would advise toxicity studies to focus on lower-level doses (2-10-fold increase in frataxin) to truly determine the lowest dose that might be toxic.

Response to Comment 4:

We agree and because of the data obtained in the current study of nonhuman primates, much lower doses of the AAV vector are being used in the current human gene therapy clinical trials in FRDA patients. We will be conducting quantitative analyses of hFXN-M in the heart tissue biopsy samples from these gene therapy trials using the method described in the current manuscript.

The text has been modified as follows:

Page 16 Line 15

'At the lowest dose of vector, no toxicity would be predicted from the observed hFXN-M levels (Fig. 7A), although longer timepoints and increased sample sizes would be needed to fully assess the possible toxicity of such hFXN-M levels.'

Specific & technical points:

Specific Point a. Calculation of the fold-increase in frataxin with differing AAV dosage might be done differently. I'm not sure but it seems you are calculating the amount increase over the endogenous frataxin. I would have used the amount of frataxin at the lowest dose as the baseline (or assign it as 1) and do fold changed for tother doses over that. In this case, my calculation is 9.5-fold at the mid-level dose and 16.75-fold at the highest dose. But as I stated in comment 2 above, these calculations will be subject to the precision of actual protein relative to a loading standard.

Answer to Specific Pont a: The text has been modified as follows:

Page 14 Line16

"hFXN-M concentrations were determined from a standard curve prepared by adding increasing amounts of hFXN-M to the same amount of SILAC hFXN-M that was added to the tissue samples. Therefore, SILAC hFXN-M in effect, acts as the loading control because the tryptic peptides are identical to those derived from hFXN-M, differing only in mass. The ratio between hFXN-M and mFXN-M expressed in mouse heart tissue to the SILAC hFXN-M internal standard is established at this time. Therefore, corrections do not have to be made for any losses during the subsequent isolation and analysis procedure. The FXN-M amounts were calculated from the ratio of hFXN-M and mFXN-M tryptic peptides to the corresponding SILAC tryptic peptides (Fig. 4 A-D and Fig. 6 A-D) using the regression lines of relevant standard curves (Figs. 2B and 2C). The standard curves were prepared at the time of the analysis using the same isolation procedure as used for the tissue samples. The precision and accuracy of these measurements are provided in Table 2."

Page 16 Line 3

"The lowest dose of AAV vector of $1.00E13$ GC/kg resulted in the expression of hFXN-M at a mean level of 4.1 ± 2.8 ng/mg tissue (Fig. 7A) which is close to the endogenous level of hFXN-M in human heart and mFXN-M in monkey heart (Fig. 4E). In contrast, a three-fold increase in dose of the AAV vector ($3.00E13$ GC/kg) resulted in an almost 10-fold increase in the level of hFXN-M protein (37.9 ± 17.6 ng/mg protein) in heart tissues compared with endogenous mFXN-M (4.1 ± 2.8 ng/mg protein) in the same tissues (Fig. 7B). Furthermore, heart tissue hFXN-M levels (67.7 ± 15.4 ng/mg protein) after a ten-fold dose increase in vector ($1.00E14$ GC/kg) were 20-fold higher than the endogenous levels of mFXN-M (2.8 ± 1.2 ng/mg protein) in the monkey heart (Fig. 7C)."

Specific Point b: On page 14, line 348 there is a statement “It is evident that mRNA data did not predict the observed protein levels”. What data do you speak of? Is it from these samples? If so, it should be shown.

Answer to Specific Pont b: The text has been modified as follows:

Page 16 Line 13

“Furthermore, even with limited sample sizes and the short 28-day study duration, it was clearly evident that the dose administered did not predict the observed hFXN-M protein levels.”

c) On page 15, line 349, the statement “no toxicity was observed” should be supported by some data.

Answer to Specific Pont c: The text has been modified in response to Specific Point c

d) In figure 1, what does the brown arrow signify?

Answer to Specific Pont d: The Figure has been amended to make it clear that this is the result of MPP-mediated cleavage to FXN-M

e) In figure 5, panels A, B, and C, seem to come from different. Why such inconsistency?

Answer to Specific Pont e: The signals In Fig. 5 are from the light peptides. Quantification was conducted by measuring the ratio of the signal from the light peptide to the signal from the heavy peptide (internal standard) as illustrated in Figs. 4 and 6. Fig. 5 was intended to show the huge increase in signals from hFXN-M light peptides after higher doses of vector (Panels B and C) when compared with endogenous mFXN-M light peptides, which are barely detectable.

f) Which parts of the heart do the samples in figure 3 come from?

Answer to Specific Point f: The text has been modified as follows:

Page 9 Line 8

“Levels of hFXN-M in five different control (non-FRDA) left ventricle human heart tissue were determined to be 5.1 ± 1.7 ng/mg tissue (mean \pm SD; n=5). Levels of mFXN-M in five different Rhesus macaque left ventricle heart tissue from untreated animals were 2.1 ± 0.4 ng/mg tissue (Fig. 4E).”

Fig. 3 Legend

“UHPLC-MR/MS analysis of FXN-M tryptic peptides (A) hFXN-M from human heart left ventricle. (B) mFXN-M from Rhesus macaque heart left ventricle.

g) In the abstract, I may have missed it but has the pathogenesis of disease in FRDA been exclusively assigned to mature frataxin? If not, it might be best to just say frataxin without such a modifier.

Answer to Specific Point g: yes

h) In the abstract “up to 1400 GAA triplet repeats” may not be entirely accurate. No upper limit has been definitively assigned to the trinucleotide repeat expansion. It might be best to say something like “from 70 to >1000 repeats”.

Answer to Specific Point h: The text has been modified as follows:

Page 2 Line 4

“It results primarily through epigenetic silencing of the *FXN* gene by GAA triplet repeats on intron 1 of both alleles. GAA repeat lengths are most commonly between 600 and 1200 but can reach 1700. A subset of approximately 3 % of FRDA patients have GAA repeats on one allele and a mutation on the other.”

Page 3 Line 14

GAA repeat lengths are most commonly between 600 and 1200 in FRDA patients¹ but a repeat length of 1700 has been reported¹².

Reviewer #2:

Overview: In this work, Rojsajakul and cols develop a method to specifically quantify the mature frataxin protein after gene therapy studies. This work covers one of the remain questions we still have in the gene therapy field regarding the amount of protein is produced from the AAV vector after treatment and how this “exogenous protein” is distinguish from the endogenous one.

Methodologically this work is brilliant, results are convincing and conclusions very well argued. However, I do still have a few points that should be addressed before accepting it:

Response to Overview: Needless to say, we are delighted with the reviewer's appraisal of our work. We hope that our responses the reviewer's points will allay any remaining concerns.

Point 1: As you shown in the results section, once low and medium dose of the vector was injected, the amount of monkey FXN (mFXN) was not affected. However, it seems there is a reduction of mFXN after the injection of the high dose of the vector was done. This could be a side effect due to the dose used or maybe a limitation in the method to quantify FXN when this protein is highly overexpressed. I wonder whether you have an explanation for this and if so, please explain it in the manuscript.

Answer to Point 1: The text has been modified as follows:

Page 12 Line 2

“There appeared to be a reduction in the amount of mFXN-M that was expressed in the heart, which might have been due to the problem of quantifying mFXN-M in the presence of such high levels of hFXN-M. However, the hFXN-M levels expressed in the heart were still 32.2-fold higher than the mFXN-M levels we found in control monkey hearts (2.1 ng/mg tissue, Fig. 4E).”

Point 2: In the Belbella study (2020) they showed toxicity in the heart when 20-fold of FXN overexpression was reached after AAV treatment. They showed this toxicity 12-months after the

treatment. In your study you do not detect any abnormality in hearts when the high dose was used, even though the amount of FXN detected was 24-fold higher. You claimed that this is due the length of your study (4 weeks). However, in Belbella study they do not detect anatomical anomalies in the heart either, but several disruptions in the heart at histological levels were detected as well as mitochondria disruptions. I wonder whether you have analysed the tissue collected from your study at histological levels. If you still have some left over cardiac tissue, this could help to detect some abnormalities even though your study was shorter in time.

Answer to Point 2: The main purpose of the study was to determine the tissue levels of the hFXN-M and mFXN-M not to conduct a toxicity study, which is on-going. Unfortunately, we used up all the samples developing the bioanalytical method. However, a much larger study is under way in which we will report the toxicity data once it is completed. We have modified the text as follows to also address the concerns of reviewer 1.

Page 16 Line 13

“Furthermore, even with limited sample sizes and the short 28-day study duration, it was clearly evident that the dose administered did not predict the observed hFXN-M protein levels. At the lowest dose of vector, no toxicity would be predicted from the observed hFXN-M levels (Fig. 7A), although longer timepoints and increased sample sizes would be needed to fully assess the possible toxicity of such hFXN-M levels.”

Point 3: Most of the gene-therapy preclinical studies are done first in mice before human non primates, do you think this method of quantification is extrapolated to other species?

Answer to Point 3

We have just completed a study in mice using similar methodology used in the present study and are in the process of preparing a manuscript describing the work. It is particularly interesting because mouse FXN is processed differently from hFXN (Ref 22). Therefore, this new study has provided the opportunity to determine whether hFXN expressed in mouse heart undergoes similar processing to mouse FXN that is expressed in the same tissues.

Pont 4: Reference 15 regarding gene-therapy drugs already approved is from 2021, however a new gene therapy treatment was approved by 2022. Please, could you find a more recent reference for this?

Answer to Point 4: The text has been modified as follows:

Page 4 Line 5

“In addition, the potential of these vectors has been established by numerous preclinical and clinical studies, as well as by already approved therapies^{17,18}”

Pont 5: There are a few typos throughout the manuscript: hFXN-M seems to be repeated, (page 4, 2nd paragraph, line 4); “tin” (page 12, line 7);”in” (page 14, lane 4)

Answer to Point 5: The typos have been corrected and the manuscript thoroughly checked for possible additional typos.

REVIEWERS' COMMENTS:

Reviewer #1 (Remarks to the Author):

The revised manuscript by authors Rojsajakul et al, titled "Quantification of human mature frataxin protein expression in nonhuman primate hearts after gene therapy" presents a point-by-point response to critiques and comments from 2 reviewers. The responses consist of clarifications of items within the text and rebuttals of the reviewer concerns with text revisions to justify their positions. There is no additional or new data to be considered. Overall, this represents a moderately responsive revision.

In its present state, the data in this manuscript shows the use of well-established state-of-the-art methods for precise measurement of the efficacy of virally mediated gene expression in the intact heart of a non-human primate. It represents a useful technological methods paper that will be of interest to scientists and pharmaceutical companies developing and may be adaptable to other genes and target tissues.

There are no major flaws of concern at this point.

Reviewer #2 (Remarks to the Author):

In their revision, Rojsajakul et al have completely addressed my comments. Even I still have certain concerns regarding some toxicity they could have been underestimated, authors claim they have a tox-study study ongoing. Therefore, under my point of view, I think this paper is ready to be published.